



# Correcting precipitation measurements of TRwS204 in the Qilian Mountains, China

Qin Zheng[1,2], Rensheng Chen[1], Chuntan Han[1], Junfeng Liu[1], Yaoxuan Song[1], Zhangwen Liu[1], Yong Yang[1], Lei Wang[1,2], Xiqiang Wang[1,2], Xiaojiao Liu[1,2], Shuhai Guo[1,2], Guohua Liu[1,2]

[1]Qilian Alpine Ecology and Hydrology Research Station, Key Laboratory of Inland River Ecohydrology, Northwest Institute of Eco-Environment and Resources, Chinese Academy of Sciences, Lanzhou 730000, China
[2]University of Chinese Academy of Sciences, Beijing 100049, China

*Correspondence to*: Rensheng Chen (crs2008@lzb.ac.cn)

**Abstract.** With the development and popularization of automatic weather stations, testing the performance of the recording

precipitation gauges and deriving the adjustment algorithm have become the top priority. This study mainly analyzed the losses of TRwS$_{SA}$ (TRwS204 shielded with a single Alter) through correlation and regression methods, and derived the correction algorithm from August 2014 to August 2016 in the Qilian Mountains, China. Results show that precipitation collected with TRwS$_{SA}$ was 116.2, 5.8, and 7.6 mm less than the "true" precipitation during the experiment for rain, sleet, and snow, respectively. For the losses, specific errors account for a larger proportion than systematic errors for rainfall and

snowfall events, while systematic errors account for a larger proportion than specific errors for sleet events. Regression analyses show that the amount of precipitation and mean air temperature can affect specific errors, particularly for snowfall events. On average, the specific errors per event were 0.6, 0.0, and 0.4 mm for rain, sleet, and snow, respectively, and the systematic errors per event were 0.1, 0.1 and 0.0 mm for rain, sleet, and snow, respectively. For systematic errors, wind speed was still the most significant factor for the catch ratio (CR) of rain and sleet, whereas humidity affected the CR of

snow to a certain extent. Currently, given that the transfer functions were agreed to derive from the DFAR (DFIR fence + automatic weighing gauge + shield + precipitation detector), considerable attention should be focused on the specific errors of the automatic weighing gauge.



## 1 Introduction

Reliable precipitation datasets play an important role in the hydrological process model and climate change research on the regional and global scales (Yang and Ohata, 2001). Precipitation datasets were mainly obtained from two basic sources, namely, the point gauge measurements and the remote sensing algorithms. However, both methods have uncertainties and biases. Generally, point precipitation measurements are chosen as a reference to calibrate the satellite precipitation algorithms. Thus, given the importance of point precipitation measurements, the biases in gauge measurements must first be evaluated.

Several factors have long been recognized to account for the biases in gauge-measured precipitation. Errors caused by these factors can be divided into two main types, namely, random and systematic errors. Previous studies show that random errors in point precipitation measurements are mainly caused by terrain, microclimate, uneven distribution of precipitation in space, and particularity of the experimental sites (Ren et al., 2003). By contrast, systematic errors in point precipitation measurements are primarily caused by aerodynamic effect, wetting loss, evaporation loss, trace precipitation, and "false" precipitation (Bogdanova et al., 2002). Given the existence of these errors, a large number of experiments and investigations, which aimed at quantifying the magnitude of these errors and further calibrating the precipitation measurements, have been implemented worldwide (e.g., Yang et al., 1988; Yang et al., 1998; Ye et al., 2004; Zhang et al., 2004; Benning and Yang, 2005; Yang et al., 2005; Ye et al., 2007; Fortin et al., 2008; Yang and Simonenko, 2014). The World Meteorological Organization (WMO) has successively organized several international precipitation measurement intercomparisons since 1955. As a result, reference standards for different phases have been proposed and correction techniques have been derived continuously (Sevruk et al., 2009). With the automatic field instruments widely operating in national weather stations, testing the performance of the automatic precipitation gauges and validating field references (with automatic instruments), which are necessary in practical work, have become the new objectives in the WMO Solid Precipitation Intercomparison Experiment (SPICE) project (Yang, 2014). Moreover, in the fourth session of the International Organization Committee for the WMO-SPICE, the transition from manual to automatic observations was highly encouraged. In the fifth session, comparisons between manual and automatic DFIR (Double Fence International Reference) observations were analyzed by Smith (2014) and Wong (2014), respectively. However, in contrast to manual precipitation gauges, automatically measured gauges not only tend to underestimate precipitation in the presence of wind, particularly snow (Fortin et al., 2008), but are also subjected to the effects of specific errors (e.g., temperature effects and software problems) (Sevruk and Chvĺla, 2005). Thus, intercomparisons at different sites around the world should be conducted to test the performance of the automatic system and correct the precipitation measurements.

According to the WMO-SPICE, the intercomparison sites could have their own specific measurement objectives, including sites with a predominance of precipitation climates and those with peculiar weather regimes. This study mainly focused on precipitation gauge measurements in semi-humid and alpine regions of China, where accurate precipitation measurement plays a significant role in scientific research and economic life. Several models (particularly OTT, MRW500, and MPS) of



automatic weighing gauges have been proven to have less relative errors and be more stable than tipping bucket gauges (Lanza et al., 2006) during the rain intensity intercomparison (2004–2008). This study aims to test the performance of TRwS204 (a total rain weighing sensor of the MPS system) and derive the correction algorithm in the Qilian Mountains, China.

## 2 Materials and Methods

### 2.1 Site and Data Sources

An experimental site (99 ° 52.9′ E, 38 ° 16.1′ N; 2,980 m) was selected on the relatively flat grassland of the Hulu watershed (upstream of Heihe) in the Qilian Mountains of the northeastern edge of the Tibetan Plateau. The climate of the experimental site is continental and alpine. The annual total precipitation varies from approximately 400 mm to 600 mm, and the annual mean air temperature (from August 2014 to August 2016) was approximately 0.7 ℃. In 2008, a meteorological cryosphere-hydrology observation system was established at the site (Chen et al., 2014). A series of related intercomparison experiments were conducted at the site to evaluate the performance of the Chinese national gauges with various configurations and obtain more accurate precipitation datasets from the Alpine regions (Chen et al., 2015). As needed, an unshielded Chinese standard precipitation gauge (CSPG; height = 70 cm and orifice area = 314 $cm^2$), a CSPG shielded with a single Alter ($CSPG_{SA}$), a CSPG in a pit, a CSPG with a Tretyakov shield and a double fence ($CSPG_{DFIR}$), and an automatic weighing gauge (TRwS204; height = 54 cm and orifice area = 200 $cm^2$) shielded with a single Alter ($TRwS_{SA}$) were installed in succession. The TRwS204 is one type of total rain weighing sensor for the MPS system with resolution of 0.001 mm. The TRwS204 can measure solid and liquid precipitation and eliminate wind vibration, as well as the unreal step change of weight and evaporation.

The precipitation dataset in this study was collected from August 8, 2014 to August 8, 2016 on a daily scale. The precipitation dataset was mainly obtained from two sources, that is, the recorded value from $TRwS_{SA}$ [Fig. 1(a)] and the manually observed value from the $CSPG_{DFIR}$ [Fig. 1(b)]. The $CSPG_{DFIR}$ was observed in strict accordance with the CMA standard (2007) and performed twice a day at 08:00 and 20:00 (Beijing time), whereas the $TRwS_{SA}$ recorded precipitation data every half an hour. $TRwS_{SA}$ measurements were converted to match the diameters of the CSPG, and the converted value was subjected to a manual quality control and filtering process. The half-hour air temperature, relative humidity, and wind speed, which were measured by the meteorological tower at this site, were also available. The types of precipitation events were mainly classified as rain, sleet, and snow by the trained observer.

### 2.2 Data Analysis

Precipitation recorded by the $TRwS_{SA}$ was compared with the "true" precipitation to investigate the performance of $TRwS_{SA}$ and correct its measurements. "True" precipitation in this study refers to the corrected value based on $CSPG_{DFIR}$



measurements. Given that evaporation losses were relatively small for most weather stations in China (Ye et al., 2007), they were not corrected in this study. Meanwhile, almost no "false" precipitation caused by storm wind was observed at this site, the measurements of CSPG$_{DFIR}$ were only corrected by wetting loss and trace precipitation. As the double fence wind shield can effectively prevent wind effect, the correction equation for CSPG$_{DFIR}$ measurements is expressed as:

$$P = P_{DFIR} + \Delta P_w + \Delta P_t, \tag{1}$$

where $P$ is the "true" precipitation, $P_{DFIR}$ is the CSPG$_{DFIR}$-measured precipitation, $\Delta P_w$ is the wetting loss, and $\Delta P_t$ is the trace precipitation. According to Yang et al. (1988), the average wetting loss per observation for the CSPG is approximately 0.32 mm. To be conservative, the wetting loss was calculated once a precipitation day, and $\Delta P_t$ was assigned a value of 0.10 mm per observation of trace precipitation.

For TRwS$_{SA}$ measurements, they were corrected by specific errors and systematic errors. The systematic errors of TRwS$_{SA}$ refer to the bias induced by aerodynamic effect. The correction equation for TRwS$_{SA}$ measurements is expressed as:

$$P = P_{TRwS} + \Delta P_s + \Delta P_a$$
$$= k(P_{TRwS} + \Delta P_s), \tag{2}$$

where $P_{TRwS}$ is the TRwS$_{SA}$-measured precipitation, $\Delta P_s$ is the specific errors, $\Delta P_a$ is the aerodynamic loss and $k$ is the aerodynamic correction factor (Allerup et al., 1997). Because TRwS$_{SA}$ and CSPG$_{SA}$ were installed at the same height with the same wind shield, the systematic errors induced by wind were assumed to be the same. To determine the value of $\Delta P_a$, the correction equation for CSPG$_{SA}$ measurements was introduced as follow.

$$P = P_{CSPG} + \Delta P_w + \Delta P_t + \Delta P_a$$
$$= k(P_{CSPG} + \Delta P_w + \Delta P_t), \tag{3}$$

where $P_{CSPG}$ is the CSPG$_{SA}$-measured precipitation. Since $P_{CSPG}$ during the experiment was available, the values of $\Delta P_a$ and $\Delta P_s$ can be calculated by combining Eqs. (1), (2) and (3).

The catch ratio (CR = $1/k$) which served as the function of environmental variables was also used during analysis, particularly of the wind speed at the gauge height of the orifice. In this study, we used the wind speed at 0.7 m because TRwS$_{SA}$ and CSPG$_{SA}$ were installed with their orifice 0.7 m above ground level. To minimize the scatter in the gauge CRs, only events measured by both gauges and in which the daily corrected precipitation based on CSPG$_{DFIR}$ measurements equals or exceeds 3 mm for rain and 1 mm for sleet and snow were used in the regression analysis.

## 3 Results and Discussion

### 3.1 Total Losses of TRwS$_{SA}$

During the experiment, 207 precipitation events (days) were measured by CSPG$_{DFIR}$. Among these precipitation events, 10 were recorded with values of 0 by TRwS$_{SA}$, and more details are shown in Table 1. Compared with the traditional manual methods, the auto-recording system incurs not only the systematic errors but also the specific errors. Sevruk and Chv la




(2005) concluded that the losses of the recording electronic weight precipitation gauges were mainly caused by temperature effects and inaccurate measurements of small amounts of precipitation, coupled with longer measuring intervals. Although new gauges and better software have been developed since then, these kinds of errors were still observed from the results. Precipitation type, air temperature, relative humidity, and wind speed were observed to have almost no relation with such a

phenomenon. Nevertheless, all these events occurred when the measured $CSPG_{DFIR}$ values were small. Given that the measuring interval was set to 30 min, these events can be explained by the combined effect of light precipitation and longer measuring intervals. Furthermore, the variable temperature caused by the complex weather in mountainous areas may also affect the normal operation of automatic weighing precipitation gauges.

However, the automatic and manual methods failed to measure 12 events of trace precipitation reported by the observer.

Given that all half-hour $TRwS_{SA}$ measurements were quality-controlled by comparing them with the measurements at adjacent sites in the Hulu watershed in the same period, this result may be explained by the complex microtopography in mountainous areas. The magnitude of this kind of random error cannot be determined because only one set of experiments was conducted. Thus, the reported 12 events of trace precipitation were not included in subsequent analysis. Related experiments are expected to be conducted at this site in the near future.

Statistical analyses of the data (Table 2) show that rainfall events dominate precipitation in the study area, followed by sleet and snowfall events. During the period of precipitation, the daily mean air temperature was 9.8 °C, 1.2 °C, and −5.7 °C for rainfall, sleet, and snowfall events, respectively. Snowfall events had clearly higher daily mean wind speed than sleet and rainfall events at the height of 0.7 m. However, all of these mean wind speeds were moderate and not more than 1.5 m s$^{-1}$. In total, precipitation collected with $TRwS_{SA}$ was 116.2, 5.8, and 7.6 mm (0.8, 0.2, and 0.4 mm per event on average) less than

the "true" precipitation for rainfall, sleet, and snowfall events, respectively. This result was unexpected because the amount of losses for rain was large during the experiment. It is clearly that the measurements of $TRwS_{SA}$ at this site should be corrected. The total measurement ratio ($TRwS_{SA}$ measurements/"true" precipitation) varied with precipitation type in the following order: $R_{sleet,total} > R_{rain,total} > R_{snow,total}$. The total measurement ratio for rain was not the highest because $TRwS_{SA}$ had more losses for rain than for sleet and snow. Even though the losses were less for snowfall events, their measurement

ratio was minimal throughout the experiment.

### 3.2 Linear Correlation of $TRwS_{SA}$ Measurements and "True" Precipitation

Figure 2a presents the scatter plot of $TRwS_{SA}$ measurements versus "true" precipitation for rainfall events. The amounts of "true" precipitation for rainfall events varied from 0.1 mm to 29.3 mm and were mostly in the range of 0 mm to 15 mm. For most rainfall events, the corrected $CSPG_{DFIR}$ measurements had been proven to be "true" precipitation throughout the

experiment. Only in a few cases did $TRwS_{SA}$ measurements marginally exceed "true" precipitation, with average overmeasurements of 0.6 mm. Further analysis showed that almost all overmeasurements of $TRwS_{SA}$ occurred in situations where moderate intensity rainfall was present at the time of manual observation. Thus, in these cases, "true" precipitation




can be underestimated. The absolute difference between $TRwS_{SA}$ measurements and "true" precipitation varied from 0 mm to 6.1 mm. Minor differences were mainly concentrated in the range of 0 mm to 4 mm, with a mean difference of 0.5 mm. For the precipitation range of 4.1 mm to 29.3 mm, the mean difference was 1.3 mm. Regression analysis revealed a close correlation between $TRwS_{SA}$ measurements and "true" precipitation for rainfall events. A linear relationship with a

5 correlation coefficient of 0.96 is statistically significantly at the 95% confidence level. This relation can be considered a transfer function to convert the $TRwS_{SA}$ measurements to "true" precipitation at this site for rainfall events.

Figure 2b presents the scatter plot of $TRwS_{SA}$ measurements versus "true" precipitation for sleet events. Sleet measurements were observed to be smaller than rainfall measurements. The amounts of "true" precipitation for sleet events varied from 0.1 mm to 9.8 mm and were mostly in the range of 0 mm to 4 mm. In contrast to rainfall events, almost half of the precipitation

measured by $TRwS_{SA}$ for sleet events was greater than the "true" precipitation. The absolute difference between precipitation measured by $TRwS_{SA}$ and "true" precipitation for sleet events varied from 0 mm to 2.5 mm. However, the difference seemed to be large at the whole sampled magnitude range of precipitation for sleet events. The correlation between precipitation measured by $TRwS_{SA}$ and "true" precipitation for sleet events is relatively lower, with a correlation coefficient $R^2$ of 0.78.

During the experiment, fewer snowfall events occurred. Figure 2c shows that the "true" precipitation of snowfall events

varied from 0.5 mm to 6.6 mm. The mean absolute difference between $TRwS_{SA}$ measurements and "true" precipitation was smaller, with a value of 0.5 mm. In the precipitation range of 0 mm to 2 mm, the difference varied from 0 mm to 1.3 mm, whereas in the precipitation range of 2.1 mm to 6.6 mm, the difference varied from 0 mm to 0.6 mm. In most cases, $TRwS_{SA}$ collected less precipitation than the "true" precipitation. A good correlation was observed between precipitation measured by $TRwS_{SA}$ and "true" precipitation for snowfall events, with a correlation coefficient $R^2$ of 0.94 at the 95% significant level.

The losses of $TRwS_{SA}$ were considered to be caused by specific and systematic errors, with their respective proportions shown in Fig. 3. During the experiment, specific errors accounted for the most $TRwS_{SA}$ losses for rain and snow, while systematic errors accounted for the most $TRwS_{SA}$ losses for sleet. This result seems to be unexpected because systematic errors have been considered to be the biggest errors of gauge measurements for snowfall events in the past researches. As presented in Table 1, several snowfall events were recorded by $TRwS_{SA}$ with zero values during the experiment. Since

$CSPG_{SA}$ measured positive values for these snowfall events, this phenomenon can be considered to be mainly caused by specific errors. In addition, fewer snowfall events occurred during the experiment. Hence, the contribution from systematic errors to the total errors of $TRwS_{SA}$ measurements for snowfall events can be small. Because the total losses for rain were greatest, this finding indicated that the specific errors were a significant problem for $TRwS_{SA}$ when recording precipitation. However, regardless of whether the errors were specific or systematic, the losses caused by each internal or external factor

were difficult to determine. This study then analyzed the relationship between these two kinds of errors and their influencing factors.





### 3.3 Specific Errors of TRwS$_{SA}$

During the experiment, the total specific errors of TRwS$_{SA}$ were 103.1 mm. Specific errors of TRwS$_{SA}$ per event on average was 0.6, 0.0, and 0.4 mm for rain, sleet, and snow, respectively. Given that specific errors can be influenced by the amount of precipitation and temperature, Fig. 4 shows the scatter plots of specific errors versus amount of precipitation and

5 temperature. For rainfall events, specific errors vary slightly with the amount of precipitation or mean temperature during the precipitation period. However, these two correlations are opposite. Specific errors increase with the amount of precipitation, but decrease with temperature. For sleet events, specific errors increase with the amount of precipitation. Almost no relationship exists between specific errors and temperature for sleet events. For snowfall events, specific errors clearly decrease with the amount of precipitation or temperature.

However, as the specific errors of TRwS$_{SA}$ are related to software problems, the regression equation cannot be simply employed as the adjusted formula. Thus, the respective mean value of specific errors for each rainfall, sleet, and snowfall event throughout the experiment was used as the value of specific errors in the correction algorithm.

The difference between the orifice areas of TRwS204 and CSPG cannot be ignored. Although the TRwS$_{SA}$ measurements have been preprocessed and converted to match the diameter of the CSPG, whether the differences in measurements are

15 affected by the size of orifice area should still be considered. As shown in Fig. 5, the mean difference between precipitation measured by Pit$_{314}$ (a pit gauge with orifice area = 314 cm$^2$) and Pit$_{500}$ (a pit gauge with orifice area = 500 cm$^2$ next to Pit$_{314}$) throughout the experiment was relatively small. On average, Pit$_{314}$ collected 0.2 and 0.1 mm more precipitation than Pit$_{500}$ for rain and snow per event, respectively, and 0.1 mm less precipitation than Pit$_{500}$ for sleet per event. This finding indicated that gauge measurements are obviously affected by the orifice area of gauges. Therefore, a slight deviation in the specific

errors calculated previously is detected.

In the final report of the seventh session for the WMO-SPICE, the transfer functions were agreed to drive from the DFAR (DFIR fence + automatic weighing gauge + shield + precipitation detector), not the DFIR or the bush gauge. TRwS$_{DFIR}$ (DFIR fence + TRwS204 + single Alter shield) was installed at this site in the summer of 2016 (see Fig. 6) to achieve a better comparison of precipitation measurements. Given that specific errors of TRwS$_{SA}$ were a significant problem at this site,

future studies should more carefully discuss this issue when deriving transfer functions from the TRwS$_{DFIR}$.

### 3.4 Systematic Errors of TRwS$_{SA}$

Throughout the experiment, the total systematic errors (aerodynamic losses) of TRwS$_{SA}$ were 26.5 mm. As calculated, average systematic errors per event were 0.1, 0.1 and 0.0 mm for rain, sleet, and snow, respectively. This result indicated that the effect of a single Alter shield was relatively good, given that previous works have shown wind speed to be the most

30 significant factor for systematic errors (Yang, 2014). Because the systematic errors of TRwS$_{SA}$ induced by wind were assumed to be the same with those of CSPG$_{SA}$, their CRs were the same. Chen et al. (2015) analyzed the relationship





between CR of CSPG$_{SA}$ and wind speed from September 2010 to April 2015 at this site. However, the wind speed at the height of 0.7 m during the period of precipitation was not completely available during their study. In this study, the analysis of CR versus wind speed at gauge height during the period of precipitation was conducted.

Cubic polynomials and exponential functions were used to determine the relationships between CR and wind speed for different precipitation types. In this section, CR was assumed to be only affected by the wind, which indicates that CR would be equal to 100% in the absence of wind. Figure 7a presents the scatter plot of CR versus wind speed for rainfall events on a daily scale. The CRs varied from 83.7% to 112.7% at wind speeds lower than 1 m s$^{-1}$ and decreased to 86.1% to 101.6% at wind speeds higher than 1 m s$^{-1}$. This finding indicated that the scatter of CRs is larger at lower wind speeds of rainfall events.

For sleet events (see Fig. 7b), the CRs varied from 80.6% to 114.2%, with wind speeds ranging from 0.4 m s$^{-1}$ to 1.3 m s$^{-1}$. Similar to rainfall events, the scatter of CRs is larger at lower wind speeds. For snowfall events (see Fig. 7c), CRs varied from 84.9% to 107.6% at wind speeds ranging from 0.5 m s$^{-1}$ to 1.5 m s$^{-1}$. Given that wind speed was low at this site, a weak downward trend was observed in the relationship between CR and wind speed for snowfall events.

The corresponding regression equations are obtained as follows:

$$CR_{rain} = 2.98W_s{}^3 - 6.88W_s{}^2 - 0.62W_s + 100; \ R^2 = 0.05, \ N = 95, \tag{4}$$

$$CR_{sleet} = -35.46W_s{}^3 + 84.10W_s{}^2 - 51.07W_s + 100; \ R^2 = 0.08, \ N = 28, \tag{5}$$

$$CR_{snow} = 100e^{-0.05W_s}; \ R^2 = 0.14, \ N = 11, \tag{6}$$

where $W_s$ is the mean wind speed (m s$^{-1}$) at the gauge height during the precipitation period and $N$ is the number of events. Equation (4) passes the p value test with $\alpha = 0.1$, but Eqs. (5) and (6) fail with $\alpha = 0.1$. This result is basically consistent with the work of Chen et al. (2015), yet a difference in the relationship between CR and wind speed was observed for sleet events. In their study, this relationship was determined to be well-fitted with the exponential function. This finding may be explained by the different acquisition and processing methods for wind speed.

Given that wind speed was not a good explanation for the CR, other variables were considered in the regression equations. With the availability of auxiliary information, wind speed, precipitation intensity, air temperature, and relative humidity were mainly considered. Interaction terms were also considered in the regression model, given that this effect may not be a simple superposition of the influence of each variable. The result showed that CR$_{rain}$ and CR$_{sleet}$ are mainly affected by wind speed; thus, Eqs. (4) and (5) are still the best-fit functions. For snowfall events, the multiple regression equation is derived as follows:

$$CR_{snow} = 1.39R_h - 1.81T_{max}W_s; \ R^2 = 0.99, \tag{7}$$

where $R_h$ is the daily mean relative humidity (%) and $T_{max}$ is the daily maximum air temperature (℃). Equation (7) passes the p value test with $\alpha = 0.01$. For snowfall events, relative humidity and the interaction between temperature and wind speed mainly affect the CR. This result was unexpected because wind speed had been proven to be the significant factor for the





variation of $CR_{snow}$ in previous studies. However, given that snowfall events in this study were not classified as wet snow and dry snow events, humidity could have significantly affected $CR_{snow}$.

The CR values can be obtained from environmental variables if the relationship algorithm can be derived in advance. In previous studies, various methods had been applied to the regression analysis, but the derived algorithm may be unstable or inconsistent unless the elimination of random error effects and other factors, such as artificial and gauge failure, was ensured. This situation can be expressed as site-specific or data-specific, which indicates that the derived algorithm may only be applied to this site or dataset. The same undercatch for different magnitudes of measurements may cause different influences on the CR value. The CR may not be appropriate to discuss the relationship between catch ability of gauges and environmental variables when small amounts of precipitation events dominate one region.

## 4 Conclusions

This study mainly analyzed the losses of $TRwS_{SA}$ and derived the correction algorithm in the Qilian Mountains from August 8, 2014 to August 8, 2016. Light precipitation coupled with longer measuring intervals has been proven to affect $TRwS_{SA}$ significantly. A complex microclimate (particularly the variable temperature) in mountainous areas seems to be a great challenge for the operation of $TRwS_{SA}$. Correlation analysis shows that the relationship between $TRwS_{SA}$ measurements and "true" precipitation (corrected $CSPG_{DFIR}$ measurements) for rainfall events is the best, which can be considered a transfer function at this site. As calculated, precipitation collected with $TRwS_{SA}$ was 116.2, 5.8, and 7.6 mm less than the "true" precipitation during the experiment for rain, sleet, and snow, respectively. Given that $TRwS_{SA}$ and $CSPG_{SA}$ were installed at the same height with the same wind shield, systematic errors of $TRwS_{SA}$ induced by wind were assumed to be similar to those of $CSPG_{SA}$, and specific errors of $TRwS_{SA}$ were defined as the difference between precipitation caught by $CSPG_{SA}$ and precipitation measured by $TRwS_{SA}$.

Among the losses of $TRwS_{SA}$, specific errors account for a larger proportion than systematic errors for rain and snow, while systematic errors account for a larger proportion than specific errors for sleet. Regression analysis shows that the amount of precipitation and mean air temperature can affect specific errors, particularly for snowfall events. For systematic errors, wind speed is still the most significant factor for the CR of rain and sleet, whereas the CR of snow is also affected by humidity to a certain extent. This finding can be explained by the fact that snowfall events were not classified by the dry–wet condition in this study. Currently, given that the transfer functions were agreed to derive from the DFAR, considerable attention should be focused on the specific errors of the automatic weighing gauge.





**Acknowledgement.** This study was supported primarily by the National Basic Research Program of China (2013CBA01806) and the National Natural Sciences Foundation of China (41671029, 91225302 and 41401040).

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

30





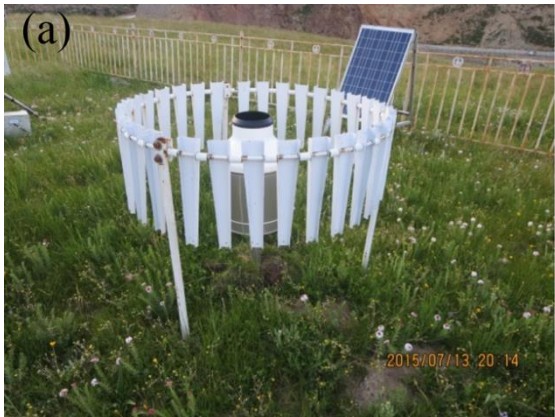
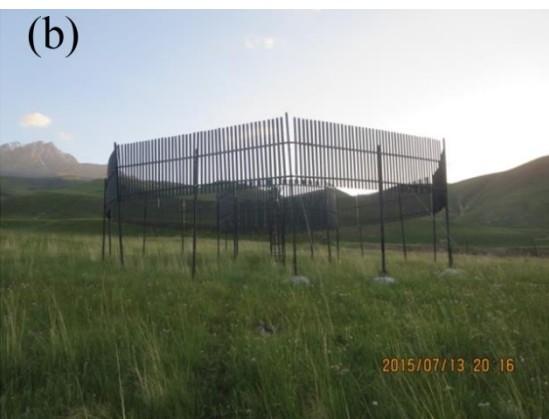

**Figure 1: The configurations of TRwS$_{SA}$ (a) and CSPG$_{DFIR}$ (b) in the Hulu watershed.**

25



**Table 1: The situation where TRwS$_{SA}$ recorded precipitation with zero values on precipitation days during the experiment.**

| Date | $P$ | Precipitation type | Air temperature | | Relative humidity | Wind speed |
|---|---|---|---|---|---|---|
| | | | Max | Min | | |
| 2014/11/1 | 1.6 | Sleet | −0.2 | −8.8 | 67.5 | 1.1 |
| 2014/11/11 | 1.0 | Snow | −3.3 | −13.0 | 76.5 | 0.8 |
| 2014/11/27 | 1.0 | Snow | −1.6 | −11.5 | 63.7 | 1.1 |
| 2014/12/20 | 0.6 | Snow | −4.2 | −17.0 | 61.5 | 0.8 |
| 2015/1/31 | 0.6 | Snow | −5.6 | −16.5 | 56.3 | 0.9 |
| 2015/2/23 | 1.3 | Snow | 2.7 | −15.4 | 60.7 | 1.3 |
| 2015/6/9 | 1.4 | Rain | 17.1 | 1.7 | 58.6 | 1.4 |
| 2015/9/5 | 0.7 | Rain | 19.8 | 1.6 | 68.3 | 0.8 |
| 2015/10/26 | 1.3 | Sleet | 6.8 | −7.0 | 65.0 | 1.0 |
| 2016/1/22 | 0.7 | Snow | −13.0 | −21.2 | 46.8 | 1.8 |

Notes: Daily mean air temperature ($\mathrm{^\circ C}$), daily mean relative humidity (%) at the height of 1.5 m, daily mean wind speed (m s$^{-1}$) at the height of 0.7 m, and $P$ refers to "true" precipitation (mm).





**Table 2: Summary of the comparison between TRwS$_{SA}$ measurements and "true" precipitation throughout the experiment.**

| Precipitation type | Events (days) | Air temperature | | | Wind speed | Total precipitation ratio | | |
| --- | --- | --- | --- | --- | --- | --- | --- | --- |
| | | Mean | Max$_{mean}$ | Min$_{mean}$ | | $P_{TRwS}$ | $P$ | |
| Rain | 152 | 9.8 | 16.5 | 4.9 | 1.0 | 785.0 | 901.2 | (mm) |
| | | | | | | 87.1 | 100.0 | (%) |
| Sleet | 35 | 1.2 | 7.5 | −3.1 | 1.1 | 99.9 | 105.7 | (mm) |
| | | | | | | 94.6 | 100.0 | (%) |
| Snow | 20 | −5.7 | 0.1 | −10.2 | 1.2 | 34.0 | 41.6 | (mm) |
| | | | | | | 81.7 | 100.0 | (%) |
| All | 207 | 6.9 | 13.4 | 2.1 | 1.0 | 918.9 | 1,048.5 | (mm) |
| | | | | | | 87.6 | 100.0 | (%) |

Notes: Daily mean, max$_{mean}$, and min$_{mean}$ air temperature (℃) at the height of 1.5 m, daily mean wind speed (m s$^{-1}$) at the height of 0.7 m, and $P_{TRwS}$ and $P$ refer to TRwS$_{SA}$ measurements and "true" precipitation, respectively.





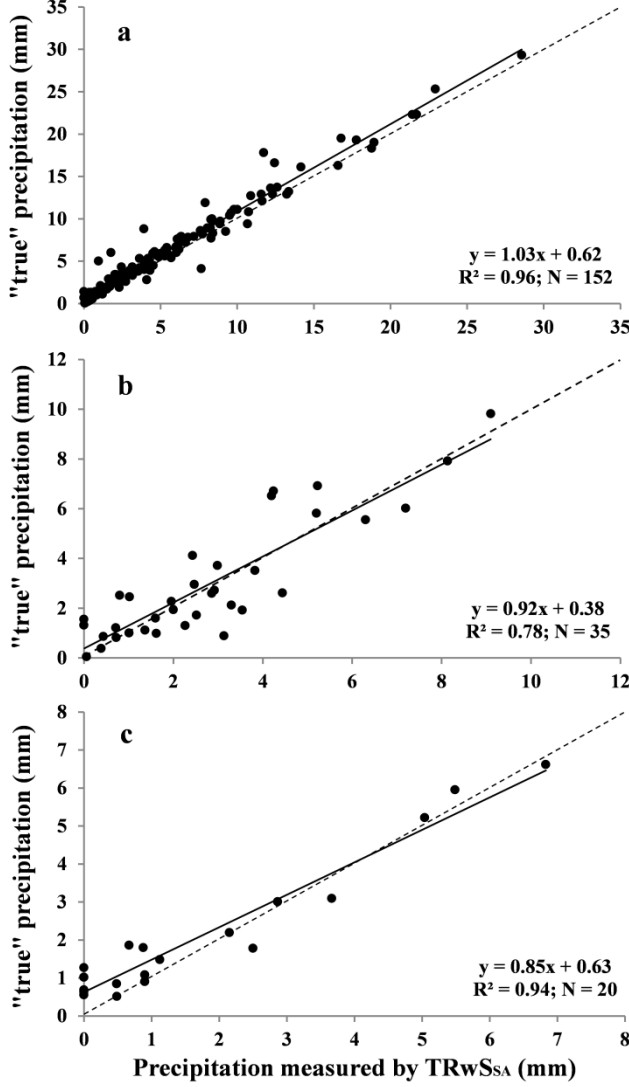

**Figure 2: Linear correlations of daily precipitation measured by TRwS$_{SA}$ and "true" precipitation for rain (a), sleet (b), and snow (c). The solid line is the fitted line, and the dotted line is the diagonal.**





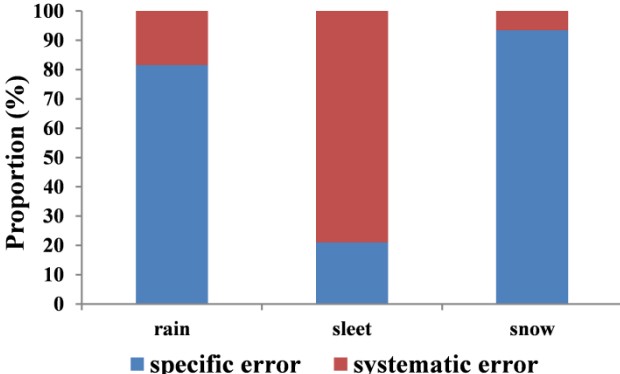

**Figure 3: Respective proportions of the specific and systematic errors for different types of precipitation throughout the experiment.**

25





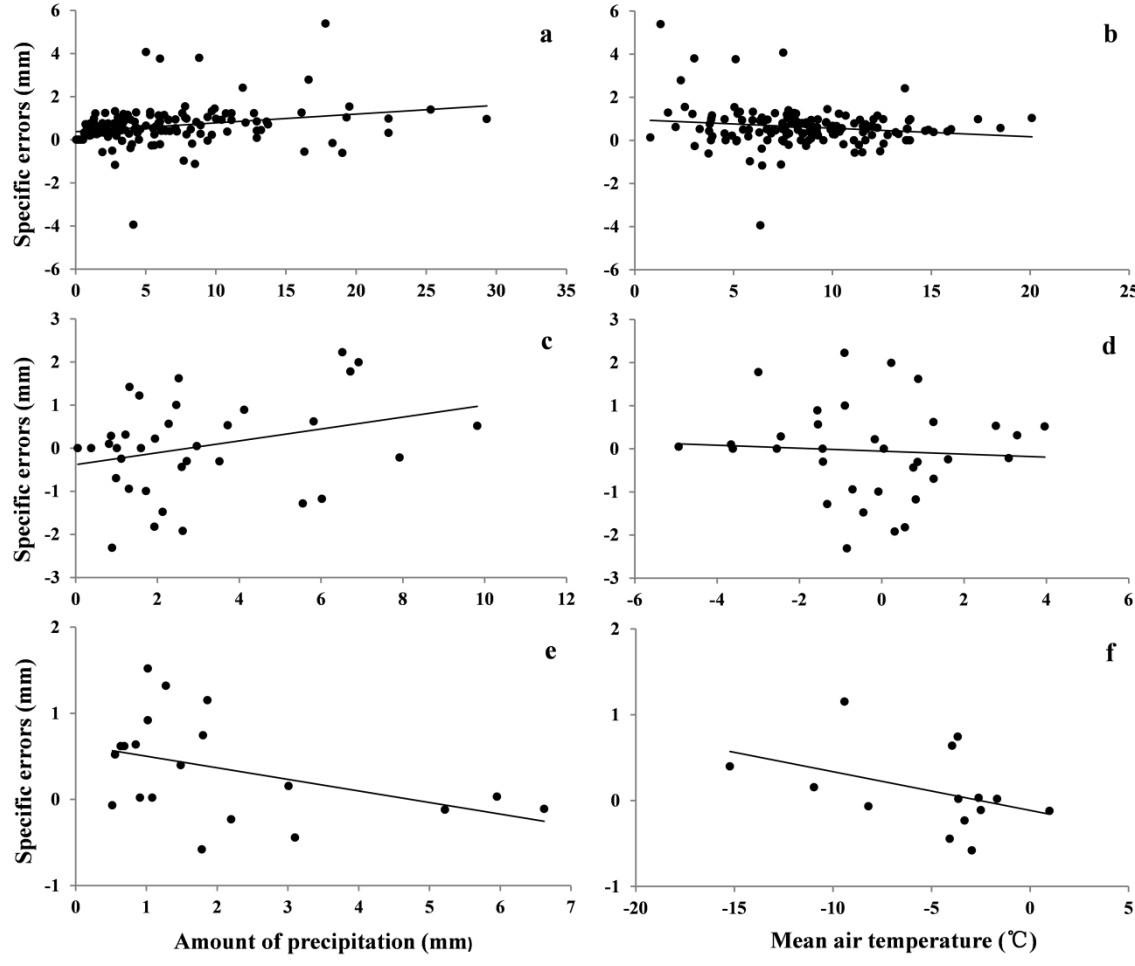

**Figure 4: Specific errors versus amount of precipitation (left) and specific errors versus mean air temperature (right) for rainfall (a, b), sleet (c, d), and snowfall (e, f) events. When discussing the relationship between specific errors and mean air temperature during the period of precipitation, events recorded by TRwS$_{SA}$ with zero values were eliminated.**





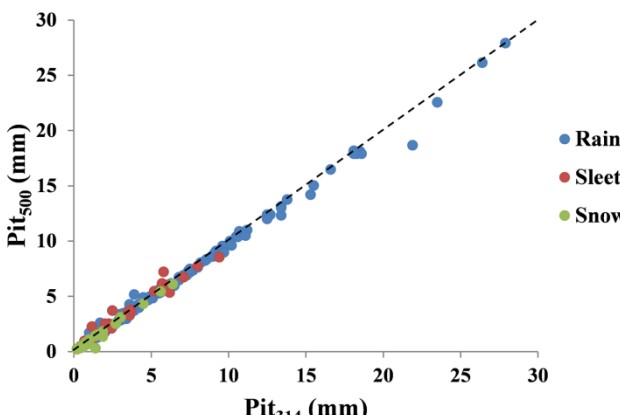

**Figure 5: Comparison between precipitation measured by Pit$_{314}$ (a pit gauge with orifice area = 314 cm$^2$) and Pit$_{500}$ (a pit gauge with orifice area = 500 cm$^2$ next to Pit$_{314}$). The dotted line is the diagonal.**

25

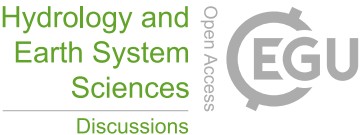



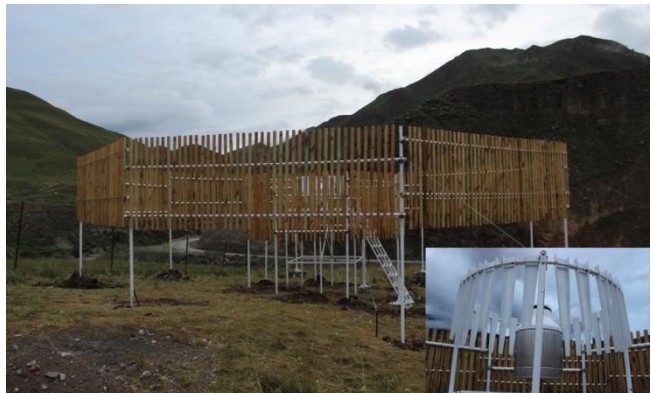

**Figure 6: The configuration of TRwS$_{DFIR}$ in the Hulu watershed.**

25

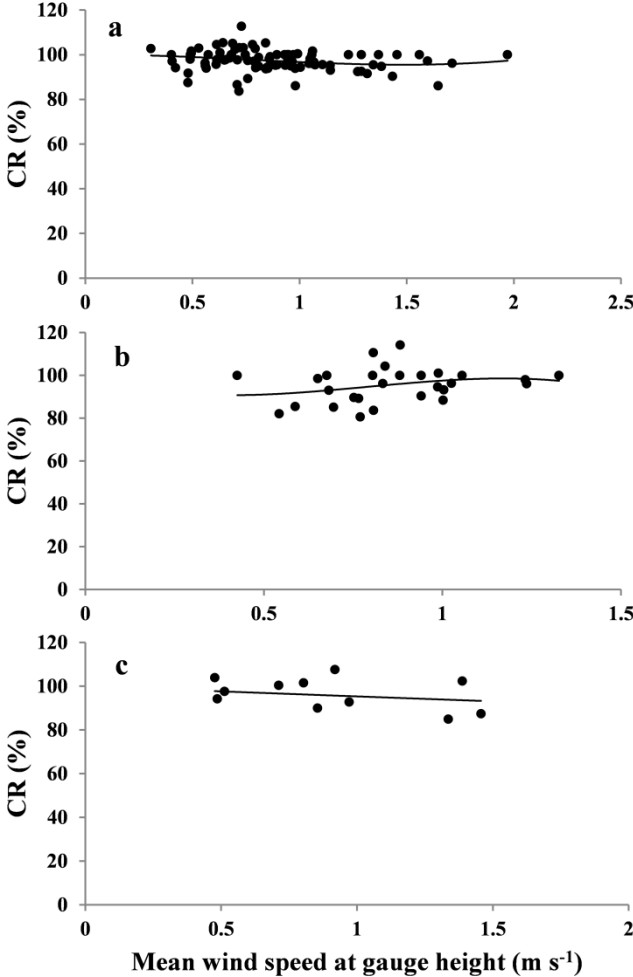

**Figure 7: CRs versus wind speed at gauge height of 0.7 m during the period of precipitation for rainfall (a), sleet (b), and snowfall(c) events.**