# Peer review of "Correcting precipitation measurements of TRwS204 in the Qilian Mountains, China"

_Hydrology and Earth System Sciences, 2017_

## Referee Comment (RC1) · J. Kochendorfer (Referee) · 11 Mar 2017

Review of, "Correcting precipitation measurements of TRwS204 in the Qilian Mountains, China".

**General comments:**

The manuscript describes adjustments for single-Alter shielded automated weighing gauge measurements derived from a two-year-long precipitation gauge intercomparison. Manual measurements within a DFIR shield

5    were compared to single-Alter shielded weighing gauge measurements to estimate errors due to wind, and manual measurements within a single-Alter shield were compared to the single-Alter shielded weighing gauge to estimate 'specific' errors experienced by the automated gauge.

**Specific comments:**

**1)** Wind speeds were relatively low at the testbed during the measurement period (a maximum of 2 m s$^{-1}$, from

10    Fig. 7), and there were only a few solid precipitation events (I count only 11 in Fig. 7c). The authors need either a longer measurement period like Chen et al. (2015), or a more thorough justification of the general applicability and usefulness of the proposed correction.

Because the purpose of the adjustments derived in the manuscript is presumably to correct gauge measurements at other sites, the adjustments must be representative of the wide range of meteorological

15    conditions that such monitoring sites may be exposed to. Such adjustments are most significant for solid precipitation occurring in high wind speeds. For example, neglecting differences in the wind speed measurement height, Chen et al. (2015) noted at this same site, "the wind speed showed no significant effect… below 3.5 m s$^{-1}$". In the manuscript under review, Fig. 2 shows that the TRwS$_{SA}$ measurements did not typically underestimate the "true" amount of precipitation, and were in fact fairly comparable to the DFIR-shielded

20    manual gauge used as a reference. This is what one would expect given the range of conditions that the site was subject to. The problem with this is that corrections derived from such measurements will not be applicable to windy monitoring sites that are subject to solid precipitation events, where such corrections are actually most significant and necessary. The measurements presented in the manuscript could be used to test some of the transfer functions available in the WMO-SPICE Special Issue and elsewhere, but they do not comprehend a wide

25    enough range of meteorological conditions for the derivation of valid and useful transfer functions.

**2)** The separation of specific and systematic/aerodynamic errors merits further examination. Based on the use of equations 1-3, two assumptions are made: 1) The single-Alter shielded CSPG is itself free from specific errors and 2) The aerodynamic error for the single-Alter shielded TRwS and CSPG are identical. While the second assumption may (or may not) be valid, the first assumption is also problematic. Comparisons of identical

30    precipitation gauges with identical shielding show that differences between like measurements in such a field site are significant. All precipitation gauge comparisons are subject to errors due to causes such as general measurement uncertainty and the spatial variability of precipitation. Such errors are not aerodynamic, but the methodology presented in the manuscript defines all differences between the two CSPG gauges as aerodynamic; there is no specific error term in Eq. 3, which indicates that the CSPG is completely free from

specific errors. A more defensible and direct way to estimate such specific errors would be to install a DFIR-shielded TRwS and compare it to the DFIR-shielded CSPG, or to simply use low wind speed or rainfall measurements, where the effects of shielding and wind are negligible.

In general I don't understand the advantages of estimating specific and systematic errors with the indirect approach described in the manuscript. Both correctable and uncorrectable errors could be examined directly by creating a transfer function, and then quantifying the remaining uncertainty in either the transfer function or the corrected measurements.

**Technical corrections:**

Pg. 1 ln. 17. It would be revealing to discuss the root mean square or the mean of the absolute values of specific errors in addition to the average.

Pg. 3 ln. 20-25. Please describe these measurements in more detail. Exactly how were the 12 hr TRwS measurements estimated from the 30-min measurements? Were there different outputs available for this gauge? For example, was the change in the absolute depth calculated or were the average 30-min intensities used? Was any smoothing or averaging performed? What type of anti-freeze and oil were used, and how was the heater configured? How were the CSPG measurements taken, by weight or by measuring stick for example? Was solid precipitation melted before being measured manually? How were the meteorological measurements recorded (heights, sensors, etc.)?

Pg. 4 ln. 10. Change 'corrected' to 'subject to'.

Pg. 4 ln. 22. "The catch ratio which served as the function of environmental…" needs to be rewritten.

Pg. 4 ln. 30-31. Manual methods are also subject to specific errors.

Pg. 5 ln. 5-7. I don't know much about the TRwS, but most weighing gauges output the total depth, allowing for small changes in the total depth to be calculated over longer time periods. This issue may be due to shortcomings in the way that the gauge measurements were logged and processed, although the stated 0.001 mm resolution seems quite good. Does this resolution translate to 0.001 mm/30-min?

Pg. 5 ln. 24-25. Please clarify, "Even though the losses were less for snowfall events, their measurement ratio was minimal throughout the experiment".

Pg. 5 ln. 29. How were the corrected $CSPG_{DFIR}$ measurements "proven" to be true in rainfall? Was it using the pit gauges?

Pg. 6 ln. 2. Clarify by rewriting, "with a mean difference within this range of 0.5 mm".

Pg. 6 ln. 20-22. Clarify how these sums were estimated. Were actual totals of the specific errors used, rather than totals of the absolute values of the specific errors? If the mean specific errors were actually significant, this indicates that there is a systematic bias between the two single-Alter gauges, and that it might be more effective to derive the $TRwS_{SA}$ adjustment by comparing it directly to the $CSPG_{DFIR}$.

5   Pg. 6 ln. 31. Clarify how the $TRwS_{SA}$ and $CSPG_{SA}$ CR were the same. Were the CR based on systematic differences, rather than measurements?

Pg. 8 ln 8. The range is not a good estimate of average errors, especially considering that there were many more measurements available at low wind speeds. Examine this statement more carefully, and try to support it by quantifying the errors (or scatter) at different wind speeds.

10   Pg. 9 ln 12. Light precipitation over longer intervals may be easier to quantify using the gauge depth, rather than the precipitation intensity.

---

## Referee Comment (RC2) · M. Wolff (Referee) · 24 Mar 2017

The paper presents the comparison of an automated weighing gauge (type TRwS 204) and the Chinese standard precipitation gauge (CSPG) inside a Double fence intercomparison reference (DFIR) with a two-year data set. Conclusions are drawn on the performance of the TRwS204.

There is known uncertainty in precipitation data. The World's Meteorological Organization (WMO)'s Solid Precipitation Intercomparison Experiment (SPICE) has tested the performance of many different gauge and wind shield configurations at several sites worldwide. The described site in the Quilian Mountains in China has a reference gauge following WMO's recommendations for a manual reference and recently added

an automated reference (not analysed in this study). Gauge comparisons from the site are valuable as they extend the knowledge about gauge performances in different configurations and locations, thus this study fits into the scope of the special issue.

The presented analysis of the two-year data set has several weaknesses and thus the drawn conclusions on the performance of the tested gauge may not be representative.

**General comments**

The presented concept of separation into systematic and instrument-specific errors (*2.2 Data Analysis and 3.3 specific errors of TRwS$_{SA}$*) is a different approach and might be interesting.
The derivation of the concept, however, is not convincing and is based on assumptions which are not further proofed. Generally, the rather small data set and very limited meteorological conditions at the site would make it difficult to evaluate another aspect of possible gauge errors in a representative way.
Instead of just stating that the wind-induced errors for TRwS$_{SA}$ and CSPG$_{SA}$ are the same, I suggest that you first present and compare data for both instruments and then eventually discuss this idea further based on the presented data and analysis.

Wind speeds during the two year measurement period reached maxixmal 2 m/s, which makes the evaluation of a wind dependent error very difficult and the derivation of an own set of wind dependent transfer functions does not make sense (*3.3 systematic errors of TRwS$_{SA}$*) as they won't be valid for other meteorological conditions or sites.
Instead, I recommend to use your valuable data sef of not yet in detail tested gauges and wind shield configurations to test existing transfer functions.

Please be aware that it is not meaningful to conclude on the effect of the single Alter wind shield (*4 conclusions*) for the light wind speeds you experienced during your measurement period.

The authors compare 30-min precipitation data of an automatic gauge with 12 or 24 hour data of a manual gauge. It remains unclear how the 12/24 hour data for the automated gauge are derived and which data product of the automated gauge is used. Beside calculated precipitation accumulations for the chosen measurement interval (here 30 min), the TRwS 204 also provides the raw bucket content. I assume that you summed up the precipitation totals of each individual 30 min period to determine the precipitation measured during the 12 h or 24 hour period? If my assumption is correct, then you divide already low daily precipitation totals as measured from the manual gauge to the 30 min periods measured by the TRwSS204 (*3.1 Total losses of TRwS$_{SA}$*). Most of those 30-min totals may be lower than the detection threshold from the gauge, thus resulting in a lower precipitation total. For those kind of comparisons, it would be more fair to either increase the measurement period of the TRwS 204 (if possible) or to use the change in the bucket content instead. The TRwS204 is a collecting gauge and over a longer period the accumulation would add up eventually, so that the accumulation is measurable.

The repeated statement of the absolute difference in precipitation measurements for different precipitation types is misguiding as it does not take into account the very different amounts measured for each precipitation type.

**Specific comments**

*1 Introduction*

[Figure]

Page 2, lines 25-26: I think you don't want to imply that only automatic gauges tend to underestimate precipitation in the presence of wind. The placement of **in contrast to** needs to be changed to make that clear.

*2.1 Site and Data Sources*

Page 3, lines 9-10: How did you determine values for annual total precipitation and annual mean air temperature? From the two years of data or any other period? Which period? Why are you using the term approximately?

Page 3, line 17: Please give a source for the stated resolution of the TRwS, I assume you took that from the provider's manual or other documentation.

Page 3, lines 17-19: Please state that the sensor comes with an in-built software which filters the data for vibration and evaporation.

Page 3, line 25: Could you please describe the filtering process and the manual quality control in some details?

Page 3, lines 25-26: Are the half-hour values averages or just aggregated values?

Page 3, section 2.1: It would be great, if you could add the following information:
- Which parameters are you recording form the TRwS?
- Map of the location of the site
- Layout of the site with instrument set up, also for anxcillary measurements like

temperature, wind, humidity, ...
- Instrument types of anxcillary measurements or citation of a more detailed description of the site
- height of the gauge inside the DFIR

Page 3, section 2.1.: I think the information on the new automated gauge (given in section 3.3.) in a double fence can also be stated here for a complete description of the site.

*2.2 Data Analysis*

Page 4, Equation 2: Please check the first line of the equation. According to Allerup et al. (1997), the corrected precipitation is
P=R$_{measured}$ $+ \Sigma(\Delta R_{errors})$=k(P$_{measured}$ $+ \Sigma(\Delta R_{errors})$)
where R, is refering to a **reference gauge** while P is refering to the gauge under test.

Page 4, Lines 15-16: For your analysis you make the assumption that the systematic errors of the CSPG$_{SA}$ and the TRwS$_{SA}$ are the same. While it is an interesting idea to separate the errors in instrument-specific and systematic error, I find that you have at this point of the manuscript to little substance to introduce this step and it seems rather artificial and complicated. I suggest that you first present and compare data for both instruments and then discuss this idea based on the analysed data.

Page 4, line 25: Earlier you introduce that manual measurements are performed twice a day (Page 3, line 23). Here you refer to daily precipitation. Are you using 12 or 24 h periods? In case of 24 h-periods (= daily) which of the two daily measurements are you using?

*3.1 Total Losses TRwS$_{SA}$*

Table 1 and related text in the section: For completeness, please add the measurements of CSPG$_{SA}$ in the table and discuss those. As discussed under general comments, I think the comparison of precipitation totals based on separate 30-min-periods with totals for 12 or 24 h periods introduce an additional (non-real) bias for the automated gauge. It is possible for the TRwS to measure the total accumulation over the longer interval, thus also the TRwS may be able to register those small amounts of preciptiation.

Further, 10 out of 207 events are less than 5%. That means, in more than 95% of the cases, the TRwS$_{SA}$ detected precipitation measured by the reference. I find that is a very important and positive result which should be mentioned. I find it doubtful to conclude on remaining instrument issues caused by temperature effects and inaccurate measurement of small amounts of precipitation from 10 single cases.

Page 5, line 9: Isn't the definition of trace that the amount of precipitation can't be measured? Thus, the fact that the manual methods did not measure the trace is redundant and not a special fact which needs to be explained by the complex microtopography of the area.

Page 5, line 18: To my knowledge are winds up to 3 m/s called light. The term moderate is used for winds higher than 5.5 m/s. That is important, because for light winds the expected wind-induced undercatch is very small.

[Figure]

Page 5, line 19: The different precipitation types occured in very different numbers (both number of events and amount of preciptation) and the comparison of absolute values is not giving the right impression on where you noticed large and small deviations. From the numbers in table 2, I calculate that the reference measured on average 2 mm snow, 3 mm sleet and 7 mm of rain per event. Your stated average undercath of 0.4 mm for snow, 0.2 mm for sleet and 0.8 mm for rain relates to 20% undercath for snow and approx. 10% undercatch for rain and sleet. I find those kind of numbers better comparable with results from other studies.

Page 5, line 20-22: The sentence "**This result was unexpected ...**" belongs into conclusions and should be discussed there.

Table 2 and related text in the section: For completeness, please add the measurements of $CSPG_{SA}$ in the table and discuss those.

*3.2 Linear Correlation of $TRwS_{SA}$ Measurements and "True" Precipitation*

Figure 2: I wonder if it is possible to combine those plots into one, using different colors for the different precipitation type, thus giving a more comparable impression of the scatter plots. For completenes, please add a similar plot(s) for the measurements of $CSPG_{SA}$. I find it questionable to calculate new regression lines, as especially for sleet and snow the low number of points make it difficult to reach statistically significance. Why not calculate the standard deviation and the standard error?

Page 6, lines 9-10: Instead of stating that half of the measured events were overestimated, I recommend to state that the deviations seems to be randomly distributed around the 1-1-line and both over- and underestimate, or generally that a larger scatter

was observed. The calculation of standard deviation would help to set this in relation to rain and snow. The different scales of the panels in figure 2 makes it difficult to see which deviations are actually "larger" and "smaller".

*3.4 Systematic Errors of TRwS$_{SA}$*

Page 7, line29: The Alter shield has also in other studies shown good effect for reducing wind-induced undercatch for light wind speeds.

Pages 8-9: I don't understand why you try to generate new transfer functions with you data set instead of applying existing (with advantage from different authors) transfer functions and discuss if they work. As you don't have data with gentle, moderate or strong wind speeds, it will be difficult to develop new representing transfer functions depending on wind speed.

---

## Author Comment (AC1) · 26 Mar 2017

**Reply to Nominated Referee #1**

We thank Dr. Kochendorfer for the comments; below we give the reply to the comments.

**Specific comments:**

**1)** Wind speeds were relatively low at the testbed during the measurement period (a maximum of 2 m s$^{-1}$, from Fig. 7), and there were only a few solid precipitation events (I count only 11 in Fig. 7c). The authors need either a longer measurement period like Chen et al. (2015), or a more thorough justification of the general applicability and usefulness of the proposed correction.

Because the purpose of the adjustments derived in the manuscript is presumably to correct gauge measurements at other sites, the adjustments must be representative of the wide range of meteorological conditions that such monitoring sites may be exposed to. Such adjustments are most significant for solid precipitation occurring in high wind speeds. For example, neglecting differences in the wind speed measurement height, Chen et al. (2015) noted at this same site, "the wind speed showed no significant effect…below 3.5 m s$^{-1}$". In the manuscript under review, Fig. 2 shows that the TRwS$_{SA}$ measurements did not typically underestimate the "true" amount of precipitation, and were in fact fairly comparable to the DFIR-shielded manual gauge used as a reference. This is what one would expect given the range of conditions that the site was subject to. The problem with this is that corrections derived from such measurements will not be applicable to windy monitoring sites that are subject to solid precipitation events, where such corrections are actually most significant and necessary. The measurements presented in the manuscript could be used to test some of the transfer functions available in the WMO-SPICE Special Issue and elsewhere, but they do not comprehend a wide enough range of meteorological conditions for the derivation of valid and useful transfer functions.

**Reply:**

| A | E | G | |
|---|---|---|---|
| TOA5 | CR1000.Std.27 | | |
| TIMESTAMP | WS_70cm_Avg | WS_1000cm_Avg | |
| TS | m/s | m/s | |
| | Avg | Avg | |
| 2016-4-14 8:30 | 1.027 | 1.777 | |
| 2016-4-14 9:00 | 0.957 | 1.767 | |
| 2016-4-14 9:30 | 0.670 | 1.143 | |
| 2016-4-14 10:00 | 0.627 | 0.933 | |
| 2016-4-14 10:30 | 1.007 | 1.333 | |
| 2016-4-14 11:00 | 1.790 | 4.070 | |
| 2016-4-14 11:30 | 1.843 | 4.840 | |
| 2016-4-14 12:00 | 1.967 | 5.147 | |
| 2016-4-14 12:30 | 1.653 | 3.743 | |
| 2016-4-14 13:00 | 1.677 | 3.967 | |
| 2016-4-14 13:30 | 2.067 | 5.240 | |
| 2016-4-14 14:00 | 2.103 | 5.513 | |
| 2016-4-14 14:30 | 2.160 | 5.630 | |
| 2016-4-14 15:00 | 1.260 | 2.677 | |
| 2016-4-14 15:30 | 4.193 | 7.470 | |
| 2016-4-14 16:00 | 4.020 | 7.367 | |
| 2016-4-14 16:30 | 5.570 | 10.390 | |
| 2016-4-14 17:00 | 3.107 | 5.737 | |
| 2016-4-14 17:30 | 1.317 | 3.360 | |
| 2016-4-14 18:00 | 0.853 | 2.227 | |
| 2016-4-14 18:30 | 1.063 | 2.563 | |
| 2016-4-14 19:00 | 0.687 | 1.910 | |
| 2016-4-14 19:30 | 0.227 | 1.607 | |
| 2016-4-14 20:00 | 0.480 | 0.977 | |
| 2016-4-14 20:30 | 0.990 | 2.150 | |

Figure 1: An example of the wind speed at this site.

In fact, the wind speeds at this site were not all low (e.g., Figure 1). However, because most half-hour wind speeds were low, the average wind speeds (especially at gauge height) during

precipitation were relatively low. It really seems difficult to derive a general transfer function which can be applied in a wide range of meteorological conditions from these data. Given this, the constructive suggestion made by Dr. Kochendorfer that the measurements presented in this study could be used to test some of the transfer function available in the WMO-SPICE Special Issue and elsewhere will be a good alternative. Additionally, the less solid precipitation events during the experiment are really a problem for the analysis work. We consider updating the experimental data to make up the deficiency as far as possible.

**Specific comments:**

**2)** The separation of specific and systematic/aerodynamic errors merits further examination. Based on the use of equations 1-3, two assumptions are made: 1) The single-Alter shielded CSPG is itself free from specific errors and 2) The aerodynamic error for the single-Alter shielded TRwS and CSPG are identical. While the second assumption may (or may not) be valid, the first assumption is also problematic. Comparisons of identical precipitation gauges with identical shielding show that differences between like measurements in such a field site are significant. All precipitation gauge comparisons are subject to errors due to causes such as general measurement uncertainty and the spatial variability of precipitation. Such errors are not aerodynamic, but the methodology presented in the manuscript defines all differences between the two CSPG gauges as aerodynamic; there is no specific error term in Eq. 3, which indicates that the CSPG is completely free from specific errors. A more defensible and direct way to estimate such specific errors would be to install a DFIR shielded TRwS and compare it to the DFIR-shielded CSPG, or to simply use low wind speed or rainfall measurements, where the effects of shielding and wind are negligible.

In general I don't understand the advantages of estimating specific and systematic errors with the indirect approach described in the manuscript. Both correctable and uncorrectable errors could be examined directly by creating a transfer function, and then quantifying the remaining uncertainty in either the transfer function or the corrected measurements.

**Reply:**

We agree with Dr. Kochendorfer that all precipitation gauges are subject to errors due to causes such as general measurement uncertainty and the spatial variability of precipitation. The general measurement uncertainty for manual method is difficult to be determined and quantified, and we also do not know the magnitude of the random errors at this site at present. Whereas, since the observer is well trained and all gauges were installed not far away to each other, these errors can be relative small compared to errors like from aerodynamic effect and software problem (especially for automatic gauges). In this case, they can be ignored. As Dr. Kochendorfer commented, we did not discuss the errors of CSPG measurements comprehensively in the methodology, and we will clarify it in the revised manuscript. Related experiment about the random errors will be carried out at this site in the near future.

In addition, what we need to explain again is that the specific errors in this manuscript mainly refer to errors caused by weighing system problem of TRwS204. As suggested above, a more defensible and direct way to estimate such specific errors can be to install a DFIR-shielded TRwS and compare it to the DFIR-shielded CSPG. In fact, we have installed $TRwS_{DFIR}$ at this site in the summer of last year. So far, there are only half a year's data which can not be used in analysis. After full consideration, we consider to use low wind speed (lower than $1 \ m \ s^{-1}$) measurements to

compute specific errors as suggested by Dr. Kochendorfer.

For the opinion from Dr. Kochendorfer that "In general I don't understand the advantages of estimating specific and systematic errors with the indirect approach described in the manuscript", I will explain from the following several aspects. Firstly, it really does not need to estimate the specific errors for weighing gauge if DFIR shielded TRwS was chosen as reference. The transfer function can be established since the undercacth is mainly caused by systematic errors. However, we mainly want to test the performance of TRwS and correct its precipitation measurements in this study. In order to achieve these goals, we need a reference which should be more representative of the real precipitation. Compared to the weighing precipitation gauges, manual measuring gauges tend to measure more precipitation when under the same configuration both in previous work (Sevruk and Chvíla, 2005) and in our work. Hence, the corrected $CSPG_{DFIR}$ measurements were regarded as "true" precipitation in this study. Once the reference has been decided, the specific errors should be discussed because they are mainly caused by the two kinds of measurement methods. Secondly, analysis of specific errors and systematic errors can help to understand the error sources comprehensively, and achieve more clearly and better correction.

**Technical corrections:**

Pg. 1 ln. 17. It would be revealing to discuss the root mean square or the mean of the absolute values of specific errors in addition to the average.

**Reply:**

We agree with Dr. Kochendorfer that "It would be revealing to discuss the root mean square or the mean of the absolute values of specific errors in addition to the average". Hence, we will consider this suggestion in our revised manuscript.

Pg. 3 ln. 20-25. Please describe these measurements in more detail. Exactly how were the 12 hr TRwS measurements estimated from the 30-min measurements? Were there different outputs available for this gauge? For example, was the change in the absolute depth calculated or were the average 30-min intensities used? Was any smoothing or averaging performed? What type of anti-freeze and oil were used, and how was the heater configured? How were the CSPG measurements taken, by weight or by measuring stick for example? Was solid precipitation melted before being measured manually? How were the meteorological measurements recorded (heights, sensors, etc.)?

**Reply:**

The TRwS is a weighing rain gauge without funnel produced by the company MPS from Slovakia. It is able to indicate the liquid as well as the solid precipitation with a resolution of 0.001 mm and an accuracy of 0.1%. Exactly, we calculated daily (24-h) measurements. The 24-h TRwS measurements were estimated from 30-min measurements through cumulative calculation. TRwS has two outputs:①Pulse output corresponds to the value of increment of precipitation; ② Opto-isolated (optional) serial output RS 485 or SDI12 standard. We used the change in the absolute depth to calculate the result. Additionally, we do not have to do any smoothing. The anti-freeze we used is ethylene glycol anti-freeze, and we did not enable heating options because the heating ring of the gauge is quite energy consuming. For the CSPG, precipitation was measured by volume for rain and sleet events, while the funnel and glass bottle were removed from the CSPG and precipitation was weighed under a windproof box for snow events (Chen et

al., 2015). Solid precipitation did not melt before measured manually. We used CR1000 datalogger (from Campbell Scientific, Inc.) to record the meteorological measurements. Wind speeds at gauge height were measured by WS200-UMB Wind Sensor (from G. Lufft Mess- und Regeltechnik GmbH ).

Pg. 4 ln. 10. Change 'corrected' to 'subject to'.
**Reply:**
Okay. We will change it in the revised manuscript.

Pg. 4 ln. 22. "The catch ratio which served as the function of environmental…" needs to be rewritten.
**Reply:**
We rewrite this sentence as "The catch ratio which can be expressed as a function of environmental variables was also used during analysis, especially of the wind speed at the gauge height of the orifice".

Pg. 4 ln. 30-31. Manual methods are also subject to specific errors.
**Reply:**
We agree with Dr. Kochendorfer that "manual methods are also subject to specific errors". In fact, the specific errors in this study mainly refer to the gauge problem for automatic weighing gauges. Nevertheless, we did not correctly interpret this view in this manuscript, and we will rewrite it in the revised manuscript.

Pg. 5 ln. 5-7. I don't know much about the TRwS, but most weighing gauges output the total depth, allowing for small changes in the total depth to be calculated over longer time periods. This issue may be due to shortcomings in the way that the gauge measurements were logged and processed, although the stated 0.001 mm resolution seems quite good. Does this resolution translate to 0.001 mm/30-min?
**Reply:**
In the study of Sevruk and Chvíla (2005), they used different weights (0.5 g, 1 g, and 4 g) to investigate the relationship between simulated precipitation amounts (0.025 mm, 0.05 mm and 0.2 mm)and different measuring intervals (1 min,2 min, 3 min, 5 min, and 10 min) for TRwS in Bohunice and Liesek. Results show that lighter simulated precipitation was more likely to be less measured over longer measurement intervals in Liesek.
Of course, this issue may also be due to shortcomings in the way that gauge measurements were logged and processed as Dr. Kochendorfer put. According to MPS-system official website introduction (http://www.mps-system.sk/pdf/Projektarbeit_final_report.pdf), the measurements can be sent to a server through a GPRS (General Packet Radio Service). The transmission of data will encounter problem if the strength of the GPRS signal is not sufficient, and some data will be lost if the strength of the GPRS signal is not sufficient during a long time.
Additionally, we did not translate this resolution to 0.001 mm/30-min.

Pg. 5 ln. 24-25. Please clarify, "Even though the losses were less for snowfall events, their measurement ratio was minimal throughout the experiment".

**Reply:**

For sleet and snowfall measurements, the losses are 5.8 mm and 7.6 mm which are significantly less than rainfall measurements (116.2 mm) during the experiment. Since the measurement ratio is expressed as the value of $TRwS_{SA}$ measurements / "true" precipitation, the less losses will produce higher measurement ratio if "true" precipitation for all precipitation types were equal. However, the total "true" precipitation for snowfall events are 41.6 mm which is significantly less than rainfall events (901.2 mm) and sleet events (105.7 mm). As calculated, the total measurement ratio for snowfall events is minimal. Since Dr. Kochendorfer put this question, we realized that maybe this sentence is not on the expression of concise. We will rewrite it in the revised manuscript.

Pg. 5 ln. 29. How were the corrected $CSPG_{DFIR}$ measurements "proven" to be true in rainfall? Was it using the pit gauges?

**Reply:**

Yes, we have compared precipitation measured by $CSPG_{DFIR}$ and $CSPG_{PIT}$ for rainfall events during experiment before choosing $CSPG_{DFIR}$ as a reference gauge. As shown below, $CSPG_{DFIR}$ performed well and it can be chosen for a reference gauge also for rainfall events in this field, neglecting the measurement uncertainty and random errors.

[Figure]

Figure 2: Precipitation measured by $CSPG_{DFIR}$ vs. $CSPG_{PIT}$ for rainfall events during experiment.

Pg. 6 ln. 2. Clarify by rewriting, "with a mean difference within this range of 0.5 mm".

**Reply:**

We change "with a mean difference of 0.5 mm" to "with a mean absolute difference of 0.5 mm within this range".

Pg. 6 ln. 20-22. Clarify how these sums were estimated. Were actual totals of the specific errors used, rather than totals of the absolute values of the specific errors? If the mean specific errors were actually significant, this indicates that there is a systematic bias between the two single-Alter gauges, and that it might be more effective to derive the $TRwS_{SA}$ adjustment by comparing it directly to the $CSPG_{DFIR}$.

**Reply:**

Yes, the losses caused respectively by specific errors and systematic errors were the actual totals of specific errors and systematic errors. Here, we think using totals of the absolute values of the

specific errors or systematic errors may be not very appropriate. Fig.3 mainly analyzes the losses of $TRwS_{SA}$ from the two contributions, and the totals of the absolute values can not explain this. We consider applying the mean of the absolute values in analysis (Pg.7 ln.3-3 and Pg.7 ln.27-28) as suggested by Dr. Kochendorfer.

We investigate the precipitation difference between $CSPG_{SA}$ and $TRwS_{SA}$ for mean wind speed at gauge height during precipitation lower than 1 m s$^{-1}$. As calculated, the mean of the absolute values of the difference is 0.78 mm, 0.89 mm and 0.40 mm for rain, sleet and snow, respectively. Since wind speed below 1 m s$^{-1}$ is relative low, the systematic bias induced by wind can be regarded as small.

Pg. 6 ln. 31. Clarify how the $TRwS_{SA}$ and $CSPG_{SA}$ CR were the same. Were the CR based on systematic differences, rather than measurements?

**Reply:**

Neglecting the different wind profile caused by these two different gauge orifice rim, gauge catchments should be the same for $TRwS_{SA}$ and $CSPG_{SA}$ because the same wind shields were used. CR refers to the value of gauge catchment / "true" precipitation. Hence, in this case, CR of $TRwS_{SA}$ and $CSPG_{SA}$ were regarded as the same. CRs were based on corrected $CSPG_{SA}$ measurements since we think that CR of $TRwS_{SA}$ and $CSPG_{SA}$ were the same. However, we realize that this assumption may not be valid because the systematic difference caused by the different wind profile can not be ignored for windy condition. Given this, applying and testing existing transfer functions can be a better choice. In this way, however, we may discuss MR (measurement ratio) instead of CR (catch ratio) because we don't know the actual catchment of $TRwS_{SA}$.

Pg. 8 ln 8. The range is not a good estimate of average errors, especially considering that there were many more measurements available at low wind speeds. Examine this statement more carefully, and try to support it by quantifying the errors (or scatter) at different wind speeds.

**Reply:**

Okay. After examining this statement by quantifying the scatter at different wind speeds, this statement is proved to be not rigorous. Therefore, we decide to delete this statement.

Pg. 9 ln 12. Light precipitation over longer intervals may be easier to quantify using the gauge depth, rather than the precipitation intensity.

**Reply:**

The statement that "Light precipitation coupled with longer measuring intervals has been proven to affect $TRwS_{SA}$ significantly" may be not sound. Although very small precipitation (eg., 0.001 mm) can be missing measured due to the not translated resolution, the total precipitation may be roughly the same. We have recognized that longer intervals may do not have much effect on the result since using the gauge depth. As mentioned above, it may be caused by the reason that the strength of the GPRS signal is not sufficient during a long time.

**Finally, thanks again for Dr. Kochendorfer's comments, which are valuable in proving the quality of our manuscript.**

**Reference:**

Chen, R., Liu, J., Kang, E., Yang, Y., Han, C., Liu, Z., Song, Y., Qing, W., and Zhu P.: Precipitation measurement intercomparison in the Qilian Mountains, north-eastern Tibetan Plateau, The Cryosphere, 9, 1995–2008, doi: 10.5194/tc-9-1995-2015, 2015.

Léonard Murisier: High resolution precipitation intensity: measurement and analysis, projektarbeit_final_report, http://www.mps-system.sk/pdf/Projektarbeit_final_report.pdf.

Sevruk, B., and Chvíla, B.: Error sources of precipitation measurements using electronic weight systems, Atmos. Res., 77, 39–47, doi:10.1016/j.atmosres.2004.10.026, 2005.

---

## Referee Comment (RC3) · J. Kochendorfer (Referee) · 27 Mar 2017

Thank you for the response,

It addresses many of my comments.

Regarding the comment made on Pg. 5 ln. 29. "How were the corrected CSPG_DFIR measurements "proven" to be true in rainfall? Was it using the pit gauges?", it would be sufficient to simply reference the similar comparison made by Chen et al., 2015 (http://www.the-cryosphere.net/9/1995/2015/).

Thanks again, John
* * *

---

## Author Comment (AC2) · 9 Apr 2017

**Reply to Nominated Referee #2**

We thank Dr. Wolff for the comments; below we give the reply to the comments.

**General comments:**

The presented concept of separation into systematic and instrument-specific errors (*2.2 Data Analysis and 3.3 specific errors of TRwS$_{SA}$*) is a different approach and might be interesting.

The derivation of the concept, however, is not convincing and is based on assumptions which are not further proofed. Generally, the rather small data set and very limited meteorological conditions at the site would make it difficult to evaluate another aspect of possible gauge errors in a representative way.

Instead of just stating that the wind-induced errors for TRwS$_{SA}$ and CSPG$_{SA}$ are the same, I suggest that you first present and compare data for both instruments and then eventually discuss this idea further based on the presented data and analysis.

Wind speeds during the two year measurement period reached maximal 2 m/s, which makes the evaluation of a wind dependent error very difficult and the derivation of an own set of wind dependent transfer functions does not make sense (*3.3 systematic errors of TRwS$_{SA}$*) as they won't be valid for other meteorological conditions or sites. Instead, I recommend to use your valuable data set of not yet in detail tested gauges and wind shield configurations to test existing transfer functions.

Please be aware that it is not meaningful to conclude on the effect of the single Alter wind shield (*4 conclusions*) for the light wind speeds you experienced during your measurement period.

**Reply:**

We agree with Dr. Wolff that it would be better to first present and compare data for both instruments and then eventually discuss this idea further based on the presented data and analysis. Since the wind profile for TRwS204 and CSPG can be a little different, it really seems like an immature method for distinguishing between systematic errors and instrumental specific errors of automatic gauge in this manuscript. Furthermore, the fact that relative small data set and limited meteorological conditions at this site also makes the matter worse. It will be more appropriate to discuss this idea at last.

Additionally, the limited meteorological conditions may make the result of derivation of meteorological variables dependent transfer function site-specific and data-specific. We agree with Dr. Wolff that using the measurements to test existing transfer functions rather than deriving a new one.

The authors compare 30-min precipitation data of an automatic gauge with 12 or 24 hour data of a manual gauge. It remains unclear how the 12/24 hour data for the automated gauge are derived and which data product of the automated gauge is used. Beside calculated precipitation accumulations for the chosen measurement interval (here 30 min), the TRwS 204 also provides the raw bucket content. I assume that you summed up the precipitation totals of each individual 30 min period to determine the precipitation measured during the 12 h or 24 hour period? If my assumption is correct, then you divide already low daily precipitation totals as measured from the manual gauge to the 30 min periods measured by the TRwS204 (*3.1 Total losses of TRwS$_{SA}$*). Most of those 30-min totals may be lower than the detection threshold from the gauge, thus resulting in a lower precipitation total. For those kind of comparisons, it would be more fair to either

increase the measurement period of the TRwS 204 (if possible) or to use the change in the bucket content instead. The TRwS204 is a collecting gauge and over a longer period the accumulation would add up eventually, so that the accumulation is measurable.

The repeated statement of the absolute difference in precipitation measurements for different precipitation types is misguiding as it does not take into account the very different amounts measured for each precipitation type.

**Reply:**

Yes, the 24-h TRwS measurements were calculated cumulatively from 30-min measurements. We used the change in the absolute depth to calculate the result. For the raw bucket content, do you mean the raw weight? For TRwS204, it provides the total weight. I' m sorry I' m not quite sure about the meaning of the sentence "If my assumption is correct, then you divide already low daily precipitation totals as measured from the manual gauge to the 30 min periods measured by the TRwS204".

Since the depth resolution of TRwS204 is 0.001 mm, even if most of those 30-min totals may be lower than the detection threshold from the gauge, less than 9.6 mm (0.001 *48*200 = 9.6) will be missing measured by TRwS204 for 200 precipitation events (days). As recommended by Dr. Wolff, increasing the measurement period of the TRwS204 or using the change in the bucket content would be more fair to achieve this comparison. Now, it seems impossible to increase the measurement interval period of the TRwS204 during the experiment. Additionally, do you mean the change in the total weight in stating "the change in the bucket content"?

Finally, as commended by Dr. Wolff, we are aware of that "the absolute difference in precipitation measurements for different precipitation types" is not an appropriate way of expressing data. We will pay attention to this detail in the revised manuscript.

**Specific comments:**

*1 Introduction*

Page 2, lines 25-26: I think you don't want to imply that only automatic gauges tend to underestimate precipitation in the presence of wind. The placement of **in contrast to** needs to be changed to make that clear.

**Reply:**

Yes, we don't want to imply that only automatic gauges tend to underestimate precipitation in the presence of wind. We will change this sentence to "However, **compared to** manual precipitation gauges, automatically measured gauges not only ……"

*2.1 Site and Data Sources*

Page 3, lines 9-10: How did you determine values for annual total precipitation and annual mean air temperature? From the two years of data or any other period? Which period? Why are you using the term approximately?

**Reply:**

In fact, the source of annual total precipitation is from Zongxing Li et al. (2015). So I used the term **approximately**. I am so sorry that I missed this reference. I will add up it in the revised manuscript. For the annual mean air temperature, the two years (from August 2014 to August 2016) data was used. Here, I misused the term **approximately** to express that 0.7 ℃ is an approximate value. In fact, it is a matter about the number of decimal places. Hence, the term

**approximately** in "and the annual mean air temperature (from August 2014 to August 2016) was **approximately** 0.7 ℃" should be deleted.

Page 3, line 17: Please give a source for the stated resolution of the TRwS, I assume you took that from the provider's manual or other documentation.

**Reply:**

The source for the stated resolution of the TRwS204 was from http://www.mps-system.sk/pdf/TRWS204_504_205_405_LC_1v03.pdf.

Page 3, lines 17-19: Please state that the sensor comes with an in-built software which filters the data for vibration and evaporation.

**Reply:**

I am sorry that we have no idea of the in-built software which filters the data for vibration and evaporation.

Page 3, line 25: Could you please describe the filtering process and the manual quality control in some details?

**Reply:**

Okay. Firstly, we removed the 30-min precipitation data for corresponding humidity less than 50%. Secondly, we went on to remove the 30-min precipitation data for corresponding sunshine duration equal to 0.5 h. For the above two steps, we reserved the continuous 30-min precipitation data.

Page 3, lines 25-26: Are the half-hour values averages or just aggregated values?

**Reply:**

The half-hour air temperature, relative humidity, and wind speed values are averages. As questioned by Dr. Wolff, this sentence may be not rigorous, and it can be more accurate by rewritten as "The half-hour **average** air temperature, relative humidity, and wind speed…… ".

Page 3, section 2.1: It would be great, if you could add the following information:
- Which parameters are you recording form the TRwS?
- Map of the location of the site
- Layout of the site with instrument set up, also for anxcillary measurements like temperature, wind, humidity, ...
- Instrument types of anxcillary measurements or citation of a more detailed description of the site
- height of the gauge inside the DFIR

**Reply:**

We agree to Dr. Wolff the above recommendations.

One minute intensity and total sum of precipitation (weight and depth) were recording from the TRwS. For the map of the location of the site, layout of the site with instrument set up and a more detailed description of the site, we will supplement this content in the revised manuscript. The gauge inside the DFIR at this site is a CSPG (height = 70 cm); and the gauge inside the DFAR is a TRwS204 (height = 54 cm).

Page 3, section 2.1.: I think the information on the new automated gauge (given in section 3.3.) in a double fence can also be stated here for a complete description of the site.

**Reply:**

We agree with Dr. Wolff, and we will give a complete description of the site including introduction to DFAR and other gauges in the revised manuscript.

*2.2 Data Analysis*

Page 4, Equation 2: Please check the first line of the equation. According to Allerup et al. (1997), the corrected precipitation is P=R*measured* + Σ(Δ*Rerrors*)=k(P*measured* + Σ(Δ*Rerrors*) where R, is refering to a **reference gauge** while P is refering to the gauge under test.

**Reply:**

We agree with the content commented by Dr. Wolff. According to Allerup et al. (1997), the correction algorithms in our manuscript would be

$$P = P_{DFIR} + \Delta P_w + \Delta P_t$$
$$P = k(P_{CSPG} + \Delta P_w + \Delta P_t)$$
$$P = k(P_{TRwS} + \Delta P_s) \ ,$$

where $P$ is the "true" precipitation, $P_{DFIR}$ is the CSPG$_{DFIR}$-measured precipitation, $P_{CSPG}$ is the CSPG$_{SA}$-measured precipitation, $P_{TRwS}$ is the TRwS$_{SA}$-measured precipitation, $\Delta P_w$ is the wetting loss of CSPG, and $\Delta P_t$ is the trace precipitation of CSPG, $\Delta P_s$ is the specific errors of TRwS$_{SA}$, and $k$ is the aerodynamic correction factor.

The equations

$$P = P_{TRwS} + \Delta P_s + \ \Delta P_a$$
$$P = P_{CSPG} + \Delta P_w + \Delta P_t + \Delta P_a$$

where $\Delta P_a$ is the aerodynamic loss for single Alter shielded gauges, is introduced primarily to calculate the specific errors of TRwS$_{SA}$. They are a little different from general correction model in the work by Allerup et al. (1997), through which we attempt to quantify the aerodynamic losses.

Page 4, Lines 15-16: For your analysis you make the assumption that the systematic errors of the CSPG$_{SA}$ and the TRwS$_{SA}$ are the same. While it is an interesting idea to separate the errors in instrument-specific and systematic error, I find that you have at this point of the manuscript to little substance to introduce this step and it seems rather artificial and complicated. I suggest that you first present and compare data for both instruments and then discuss this idea based on the analysed data.

**Reply:**

We agree with Dr. Wolff. We did briefly describe the separation method, but made it look complicated. We will consider carefully the suggestion made by Dr. Wolff.

Page 4, line 25: Earlier you introduce that manual measurements are performed twice a day (Page 3, line 23). Here you refer to daily precipitation. Are you using 12 or 24 h periods? In case of 24 h-periods (= daily) which of the two daily measurements are you using?

**Reply:**

Yes, we used 24 h periods. As stated in page 3 line 23, manual measurements were performed twice a day at 08:00 and 20:00 (Beijing time). We used these two measurements in one day to

calculate the manual measurements of the day.

*3.1 Total Losses TRwS$_{SA}$*

Table 1 and related text in the section: For completeness, please add the measurements of CSPG$_{SA}$ in the table and discuss those. As discussed under general comments, I think the comparison of precipitation totals based on separate 30-minperiods with totals for 12 or 24 h periods introduce an additional (non-real) bias for the automated gauge. It is possible for the TRwS to measure the total accumulation over the longer interval, thus also the TRwS may be able to register those small amounts of precipitation.

**Reply:**

We will consider the suggestion made by Dr. Wolff that "For completeness, please add the measurements of CSPG$_{SA}$ in the table and discuss those". Additionally, we agree with the comment made by Dr. Wolff that "the comparison of precipitation totals based on separate 30-min-periods with totals for 24 h periods introduce an additional (non-real) bias for the automatic gauge". It may cause the fact that the small amounts of precipitation were failed to be recorded. However, this undercatch would be small (less than 0.048 mm/24 h).

Further, 10 out of 207 events are less than 5%. That means, in more than 95% of the cases, the TRwS$_{SA}$ detected precipitation measured by the reference. I find that is a very important and positive result which should be mentioned. I find it doubtful to conclude on remaining instrument issues caused by temperature effects and inaccurate measurement of small amounts of precipitation from 10 single cases.

**Reply:**

We agree with Dr. Wolff that the fact that TRwS$_{SA}$ detected precipitation measured by the reference in more than 95% of the cases is a very important and positive result. It should be mentioned while we overlooked this point. We will state it in the revised manuscript. Additionally, we did somewhat hurriedly came to the conclusion that "Given that the measuring interval was set to 30 min, these events can be explained by the combined effect of the light precipitation and longer measuring intervals" which need to be given further consideration. This issue may also be due to shortcomings in the way that gauge measurements were logged and processed. According to MPS-system official website introduction (http://www.mps-system.sk/pdf/Projektarbeit_final_report.pdf), the measurements can be sent to a server through a GPRS (General Packet Radio Service). The transmission of data will encounter problem if the strength of the GPRS signal is not sufficient, and some data will be lost if the strength of the GPRS signal is not sufficient during a long time.

Page 5, line 9: Isn't the definition of trace that the amount of precipitation can't be measured? Thus, the fact that the manual methods did not measure the trace is redundant and not a special fact which needs to be explained by the complex microtopography of the area.

**Reply:**

Yes, the definition of trace is that the amount of precipitation can't be measured. Here, we are mainly trying to explain the situation that the TRwS also failed to measure the 12 events of trace precipitation reported by the observer. Since the depth resolution of TRwS is 0.001 mm, it is expected to have the ability to record trace precipitation which usually can not be measured by

manual method with a resolution of 0.1 mm.

Page 5, line 18: To my knowledge are winds up to 3 m/s called light. The term moderate is used for winds higher than 5.5 m/s. That is important, because for light winds the expected wind-induced undercatch is very small.
**Reply:**
We agree with Dr. Wolff that the term "moderate" is inappropriate here. It should be replaced by the term "light" more accurately. We are sorry to have failed to consider these words carefully.

Page 5, line 19: The different precipitation types occured in very different numbers (both number of events and amount of precipitation) and the comparison of absolute values is not giving the right impression on where you noticed large and small deviations. From the numbers in table 2, I calculate that the reference measured on average 2 mm snow, 3 mm sleet and 7 mm of rain per event. Your stated average undercath of 0.4 mm for snow, 0.2 mm for sleet and 0.8 mm for rain relates to 20% undercath for snow and approx. 10% undercatch for rain and sleet. I find those kind of numbers better comparable with results from other studies.
**Reply:**
We agree with Dr. Wolff, and we are aware of this problem. We will pay attention to the way of expressing data in the revised manuscript.

Page 5, line 20-22: The sentence "**This result was unexpected ...**" belongs into conclusions and should be discussed there.
**Reply:**
OK, we will consider this suggestion in our revised manuscript.

Table 2 and related text in the section: For completeness, please add the measurements of $CSPG_{SA}$ in the table and discuss those.
**Reply:**
OK, we will add the measurements of $CSPG_{SA}$ in Table 2 and discuss those as suggested by Dr. Wolff.

*3.2 Linear Correlation of $TRwS_{SA}$ Measurements and "True" Precipitation*
Figure 2: I wonder if it is possible to combine those plots into one, using different colors for the different precipitation type, thus giving a more comparable impression of the scatter plots. For completenes, please add a similar plot(s) for the measurements of $CSPG_{SA}$. I find it questionable to calculate new regression lines, as especially for sleet and snow the low number of points make it difficult to reach statistically significance. Why not calculate the standard deviation and the standard error?
**Reply:**
Yes, we can combine those plots into one, using different colors or the different precipitation type. Since it could achieve a better comparison, we will redraw a new one. Moreover, a similar plot for the measurements of $CSPG_{SA}$ will be added as suggested by Dr. Wolff.
In addition, there can be some problems for calculating new regression lines when the precipitation events are fewer. Although this situation may be improved by extending the

experiment period, we will also try to calculate and analyze the standard deviation and the standard error.

Page 6, lines 9-10: Instead of stating that half of the measured events were overestimated, I recommend to state that the deviations seems to be randomly distributed around the 1-1-line and both over- and underestimate, or generally that a larger scatter was observed. The calculation of standard deviation would help to set this in relation to rain and snow. The different scales of the panels in figure 2 makes it difficult to see which deviations are actually "larger" and "smaller".

**Reply:**

We agree with Dr. Wolff that it will be better to state that the deviation seems to be randomly distributed around the 1-1 line and both over-and underestimate rather than state that half of the measured events were overestimated. We will adopt the suggestion made by Dr. Wolff. Additionally, the different scales of the panels in Figure 2 did make it difficult to see which deviations are actually "larger" and "smaller". As recommended by Dr. Wolff, it will give a more comparison impression of the scatter plots when combine plots in Figure 2 into one.

*3.4 Systematic Errors of TRwS$_{SA}$*

Page 7, line29: The Alter shield has also in other studies shown good effect for reducing wind-induced undercatch for light wind speeds.

**Reply:**

We agree with Dr. Wolff that the Alter shield has also shown good effect for reducing wind-induced undercatch for light wind speeds. This sentence may be described as not too rigorous and we will reconsider it.

Pages 8-9: I don't understand why you try to generate new transfer functions with you data set instead of applying existing (with advantage from different authors) transfer functions and discuss if they work. As you don't have data with gentle, moderate or strong wind speeds, it will be difficult to develop new representing transfer functions depending on wind speed.

**Reply:**

We agree with Dr. Wolff that it will be better to apply and test the existing transfer functions given the current precipitation data and limited meteorological conditions. We will adopt this suggestion in the revised manuscript.

**Finally, thanks again for Dr. Wolff's comments, which are valuable in proving the quality of our manuscript.**

**Reference:**

Allerup, P., Madsen, H., and Vejen, F.: A comprehensive model for correcting point precipitation, Hydrol. Res., 28, 1–20, 1997.

Léonard Murisier: High resolution precipitation intensity: measurement and analysis, projektarbeit_final_report, http://www.mps-system.sk/pdf/Projektarbeit_final_report.pdf.

Zongxing Li, Yan Gao, Yamin Wang, Yanhui Pan, Jianguo Li, Aifang Chen, Tingting Wang, Chuntan Han, Yaoxuan Song, Theakstone W.H.: Can monsoon moisture arrive in the Qilian Mountains in summer, Quatern. Int., 358, 113–125, 2015.

---

## Editor Comment (EC1) · M. E. Earle (Editor) · 27 Jul 2017

The authors have changed the content and structure of the manuscript significantly to address points raised by the referees, and they are commended for their efforts. An assessment of an automatic weighing gauge for precipitation measurements is presented relative to a manual reference configuration, and the comparison results (namely, the catch ratio for 12-h periods with precipitation) are used to test an adjustment function derived using the WMO-SPICE dataset. Errors specific to the weighing gauge under test are also investigated using a manual gauge in the same shield configuration. The results and analysis presented employ a dataset from a region not otherwise represented in WMO-SPICE, and are of particular interest to meteorological operations in China.

The manuscript flows logically from section to section, and is more conservative with respect to the attribution and quantification of errors than the previous submission. Questions related to the content and presentation of results are provided in the Comments section below, along with recommendations for changes/additions to the manuscript. A non-exhaustive list of proposed technical revisions is also provided. The intention of this Editor Comment is to complement the forthcoming Referee Comments, providing an additional perspective for the authors to consider as they prepare the next version of the manuscript.

**Comments**

**Title**

1. Propose changing title to 'Correcting precipitation measurements from MPS TRwS204 automatic weighing gauges in the Qilian Mountains, China'

**Abstract**

1. The abstract should be brief, but still needs to introduce necessary background material. The gauges included in the study must be introduced (e.g. MPS TRwS204 is an automatic weighing gauge, CSPG is the manual Chinese Standard Precipitation Gauge) and the 'existing adjustment function' should be elaborated upon. Further, it is stated that 'deriving adjustment algorithms has become a top priority,' but there is no mention of what these algorithms are adjusting for (e.g. wind-induced undercatch of precipitation). The abstract should be revised to include the above points.

2. When considering the results after adjustment, the following statements are made: 'It seems that the adjustment function is more appropriate to correct the snowfall measurements than rainfall and sleet measurements for this dataset.' This makes sense given the results presented, but only considers the average loss relative to the reference. What about the Root Mean Square Error? What about the Bias? It is stated that 'Overall, the results of the correction are not ideal,' but this statement is based only on the average loss, which could be impacted by a small number of events with larger losses relative to the reference. The assessment approach should be expanded, as will be discussed further in subsequent comments.

3. It is stated that 'so many factors seem to affect the differences between measurements,' but only two factors are noted (orifice area and wind profile). Other contributing factors should be described, or this sentence should be reformulated.

4. The final sentence in the Abstract would be much stronger and more broadly applicable as 'These types of errors must be considered when correcting precipitation measurement errors for different gauge types and configurations.'

**A) Introduction**

1. What is meant by "false" precipitation? (P2, L12)

2. It is stated that 'the transition from manual to automatic measurements was highly encouraged' by the SPICE IOC (P2, L25-26) – can you please elaborate on this? If I recall correctly, SPICE was organized in response to the transition to automation, not to advance or recommend this transition.

3. The Introduction transitions abruptly from a discussion of errors and adjustment functions to a discussion of manual vs. automated measurements (P2, L25-32). This information is valuable, but seems out of place here. The biases in gauge measurements are assessed relative to reference measurements; historically (e.g. the first WMO Solid Precipitation Intercomparison), the reference measurements were manual measurements using the DFIR. I suggest that the authors revise the Introduction to first describe these biases (they are presently introduced without context), then describe known biases (e.g. for automated vs. manual measurements), and then get into the different errors/contributing factors and adjustment functions. In short, the Introduction should establish the context for interpreting the results that will be presented, and should flow logically from topic to topic.

4. The transition from a discussion of biases between automatic and manual measurements to the statement 'Thus, intercomparisons at different sites around the world should be conducted to the test the performance of the automatic system and correct the precipitation measurements' is confusing. Why would testing at different sites around the world be helpful? It is difficult to follow the logic of this section, as currently presented.

5. While it is true that the SPICE intercomparison sites could have their own measurement objectives, it is not necessary to state this here (P3, L1-2), as the Qilian Mountains site was not a formal intercomparison site.

6. Which existing adjustment function? (P3, L7-8)

**B) Materials and Methods**

1. Is the DFAR configuration (with TRwS204 gauge) at the Qilian Mountains site used in the analysis? If not – which I believe to be the case – the DFAR configuration does not need to be introduced and discussed on P3, L23-27 (i.e. these two sentences can be deleted).

2. In Figure 1, it appears that the single-Alter shield slats on the $TRwS_{SA}$ (Figure 1b) are installed differently than those on the $CSPG_{SA}$ (Figure 1c). Those on the $TRwS_{SA}$ are oriented with the flat side of the slat toward the gauge (correct), while those on the $CSPG_{SA}$ are oriented with the flat side of the slat away from the gauge (incorrect). The two shield configurations are therefore not identical. The location of the centre of mass and distribution of the slat surface area will be different in each case, impacting how the slats respond to a given wind speed. The shields being identical is an important assumption in the assessment, so this difference should be noted in the manuscript.

3. How were the $TRwS_{SA}$ measurements adjusted to match the diameter of the CSPG? You indicate that the manufacturer changed a setting (P4, L10), but additional details would be helpful.

4. In the manual quality control and filtering process, are you referring to the mean 30 min humidity? The details should be provided to guide those who may want to use a similar procedure. Also, it makes sense to remove precipitation during clear sky periods (sunshine duration = 0.5 h), but the threshold duration value is not very strict. For example, if the sunshine duration was 0.49 h, would the precipitation data be included in the analysis?

5. It is stated that $TRwS_{SA}$ precipitation 'was compared with the reference precipitation to investigate the performance of the $TRwS_{SA}$ and correct its measurements.' This makes it sound like the comparison with the reference corrects the $TRwS_{SA}$ measurements, which is not the case. Perhaps it would be clearer to state simply that the $TRwS_{SA}$ measurements were assessed relative to reference precipitation measurements from the $CSPG_{DFIR}$? I don't know if it is necessary to mention the adjusted/corrected measurements and their assessment at this point.

6. Again, no information is provided to indicate what is meant by "false" precipitation (P4, L30).

7. The reasons for not deriving transfer/adjustment functions from the experimental dataset and motivation for using the transfer function developed by Kochendorfer et al. are not clearly articulated. It is stated that 'it seems difficult to derive a valid and robust transfer function for $TRwS_{SA}$ using the dataset at this site during the experimental period.' Why is this difficult? What are the limitations of the dataset? Several important points were raised during the previous review stage and discussion, which should be reflected in the manuscript.

**C) Results and Discussion**

1. Numerous (20) 12-h precipitation events were noted in which the $TRwS_{SA}$ did not report precipitation, but the $CSPG_{DFIR}$ reported precipitation. Four events were rain and sleet, during which the conditions were 'nothing special'; can you please reword and elaborate on this? 16 events were snowfall events, which were evidently characterized by lower temperatures. Do you have a theory to explain why this may have been the case? What were the characteristic wind speeds? For example, if the precipitation was light and the wind speeds were higher, it would not be surprising if the single-Alter shielded gauge missed the event. Did the $CSPG_{SA}$ report precipitation during these events?

Another important concern is whether the 12-h conditions are representative of the conditions during which precipitation actually occurred during a given 12-h period. There's not necessarily a better way that you could have addressed the conditions, but the representativeness of conditions is an important point that must be noted.

2. When discussing precipitation losses (e.g. P6, L9), it is important to specify what the losses are relative to (i.e. the reference configuration).

3. Snow accumulating on the orifice and sublimating is proposed as a loss mechanism for the TRwS$_{SA}$. Were any incidents observed during the experimental period in which snow accumulated on the orifice, or is this just a theory? Any accumulated snow could also prevent incident snowfall from entering the orifice and being measured – that is, capping of the gauge may occur – which would influence the assessment.

I would assume that the CSPG is not heated, either. Did you see any snow accumulation on the CSPG? If so, is it included in the manual measurement, or is it removed prior to the measurement? Any differences in how precipitation accumulated on the gauge orifice is dealt with for the different gauge types could potentially impact the assessment.

Is there any other reason why accumulation/sublimation would be an issue for the TRwS$_{SA}$ and not the CSPG gauges? It is stated that accumulation/sublimation 'may explain most of the situations in which the TRwS$_{SA}$ recorded values of 0 for snowfall events,' but the reasons why this does not occur for the CSPG gauges are not discussed.

4. Is data loss considered to be another precipitation loss mechanism for the TRwS gauge? If the gauge reports are based on the weight of accumulated precipitation, won't the reports following any periods of data loss include any precipitation accumulated in the bucket during the loss period?

5. The title of Table 1 indicates a comparison of events on a 12-h scale; however, the notes indicate daily (24-h) mean wind speeds and temperatures are presented. Is this correct? As noted above, the conditions during 12-h periods are not necessarily representative of the conditions during which precipitation occurred. This issue will be more significant if conditions over 24-h periods are used.

6. I would consider removing the statement 'It is obvious that the CSPG$_{SA}$ performed better than the TRwS$_{SA}$ during the experiment at this site' (P6, L32). This is a subjective statement. Alternatively, it could be changed to something like, 'It is evident that the CSPG$_{SA}$ collected more precipitation relative to the TRwS$_{SA}$ during the experiment at this site.'

At the low mean wind speeds experienced at the site over the experiment (Table 1), the differences between measurements from a single-Alter shielded gauge and those from a DFIR-shielded gauge may not be significant. In this case, the CSPG$_{SA}$ and CSPG$_{DFIR}$ measurements would be expected to be very similar, and any differences relative to the TRwS$_{SA}$ could be attributed to systematic differences in the sampling, principle of operation, aerodynamic profile, etc. between the gauge types. These systematic differences are addressed in the TRwS$_{SA}$/CSPG$_{SA}$ comparison.

7. It is stated that 'the performance of the $CSPG_{SA}$ was more stable than that of the $TRwS_{SA}$' (P7, L13-14), but this comparison does not acknowledge that the CSPG is a manual gauge (how can it be unstable?), or define what is meant by 'stable.' Is this statement made in reference to the lower standard deviation for the $CSPG_{SA}$? If so, when considering measurements over such long time scales (12-h), does the standard deviation really say anything about noise or signal variability?

8. Aside from wind speed and temperature, which other 'meteorological variables have relationships with the catch ratio' (P7, L17)? Also, when noting that air temperature and wind speed are 'typically' applied in adjustment functions, it would be valuable to include supporting references.

9. The trends described for the catch ratio as a function of wind speed and temperature in Figure 3 (text on P7, L22-24) are difficult to observe in the plots. Binning the results for each precipitation type (rain, sleet, snow) and plotting as box and whisker plots would provide a much clearer representation of the data trends, and are recommended to complement the scatter plots provided in Figure 3.

10. RMSE values are computed for observed catch ratios relative to adjusted values. It would be valuable to also include RMSE values for observed $TRwS_{SA}$ accumulation values relative to the adjusted values.

11. On page 8, the performance of the adjustment functions is assessed in terms of average precipitation losses in each precipitation type. This assessment should be complemented by RMSE and bias values for the observed $TRwS_{SA}$ accumulation values relative to the adjusted values. In the work of Kochendorfer et al., it was found that the adjustments improved the bias in value relative to the reference, and had less of an impact on the RMSE; it would be interesting to see if similar trends are observed using this experimental dataset.

12. Can you please elaborate on the 'other possible errors' that may contribute to differences between $CSPG_{DFIR}$ and $TRwS_{SA}$ measurements (P9, L4-5)?

13. Do you think that the similarity of the mean absolute differences between the $CSPG_{SA}$ and $TRwS_{SA}$ in all precip types (P9, L10) may indicate a systematic difference in measurements between these gauge configurations? That is, does the combined influence of errors specific to the $TRwS_{SA}$ result in a systematic offset in measurements relative to the CSPG?

14. In Figure 5, the cumulative sums of precipitation accumulations are plotted as a function of event number – this is effectively a time series of the total accumulated precipitation for each gauge at the end of each event. It would be far more instructive to plot the individual event accumulations (accumulation at end of 12 h period minus the accumulation at the start of that 12 h period) for the $TRwS_{SA}$ vs. those for the $CSPG_{SA}$ (or vice versa) as a scatter plot. That way, each event can be compared independently, irrespective of the precipitation accumulated in previous events. A 1:1 line (indicating perfect agreement between the gauges) can be added to illustrate the accumulation trends.

15. As presented, the results in Figure 6 are difficult to interpret. I recommend generating histograms of the differences between the $CSPG_{SA}$ and $TRwS_{SA}$ measurements for each precipitation type to more

clearly illustrate the magnitude of differences between the gauges, and how those differences are distributed. Alternatively, the TRwS$_{SA}$ vs. CSPG$_{SA}$ scatter plots (see comment above) could be modified to include only points events with mean wind speeds < 1 m/s.

16. As noted above, the TRwS$_{SA}$ and CSPG$_{SA}$ technically have the same shields, but the orientation of the slats is different (Fig. 1). This potentially complicating factor should be noted in the relevant discussion in Section 3.3.

17. It is interesting that limiting the CSPG$_{SA}$/TRwS$_{SA}$ comparison to wind speeds below 1 m/s results in almost the same mean absolute differences between the measured values as observed for the full wind speed range. I wonder if this is a reflection of the full wind speed range being low (at least in terms of the mean wind speeds in Table 1), a reflection of the mean wind speeds for 12-h periods not being representative, or an indication of differences/errors not related to the aerodynamic profile, as proposed.

18. The trends indicated in Figure 7 do not appear to be conclusive for sleet and snow, given the significant scatter and small numbers of events relative to rain (I expect the R$^2$ correlation values would be low). This caveat should be noted when making the statement, 'it may be inferred that the amount of precipitation mainly affects the specific errors of TRwS$_{SA}$ during the experiment at this site.'

**D) Conclusions**

1. As noted above, I don't think that the standard deviation of losses is a representation of measurement stability; the losses are larger for the TRwS$_{SA}$, so the standard deviation of loss values will also be larger.

2. How is it 'clear' that the 'worse correction for TRwS$_{SA}$ measurements should not be attributed to the limited meteorological conditions and precipitation data during the experiment'? I don't believe that these points were addressed in the manuscript; an earlier comment requested more details in this regard.

3. I don't know if you are in a position to say that the adjustments are 'better' or 'worse' when the changes in loss values after adjustment for both gauges are within 0.2 mm for all precip types. You indicate that the measurement uncertainty is ignored (P10, L27), but what is the estimated uncertainty for each gauge type? Perhaps more important, you are using wind speeds over 12-h periods for the adjustment that do not necessarily reflect the conditions during which precipitation actually occurred, which will impart significant uncertainty on the adjusted values. I think it is OK to state the results obtained, but any broader application of these results should be treated with extreme caution.

4. The last two sentences on page 11 are very difficult to follow. Can you please rephrase these, including relevant results from the study, if possible, to demonstrate your points?

*Proposed technical revisions*

**Abstract**

P1, L10: change to 'deriving adjustment algorithms'

P1, L10: change to 'have become a top priority.'

P1, L10: remove 'mainly'

P1, L10-13: propose restructuring as 'This study analyzed precipitation measurements from single-Alter shielded MPS TRwS204 (TRwS$_{SA}$) automatic weighing gauges relative to corrected manual measurements from the Chinese Standard Precipitation Gauge in a DFIR-shield (CSPG$_{DFIR}$) in the Qilian Mountains, China. Results were compared over the period from August 2014 to April 2017, and show that precipitation collected with the TRwS204 was…'

P1, L13-14: propose restructuring as 'Applying the adjustment function reduced the average loss of the TRwS204 by 0.2 mm for snowfall events, but increased the loss by 0.2 mm for both…'

P1, L19: delete 'also'; change 'discussion' to 'investigation'

P1, L19-21: propose changing to 'Differences between precipitation measurements from the TRwS$_{SA}$ and CSPG in a single-Alter shield (CSPG$_{SA}$) were believed to result from differences in gauge orifice areas and gauge wind profiles.'

**1. Introduction**

P2, L1: 'Accurate and reliable precipitation datasets'

P2, L2: change 'were mainly' to 'are typically' and delete 'basic'

P2, L12: '…caused by aerodynamic effects, wetting losses, evaporation losses, trace precipitation…'

P2, L17: delete 'successively'

P2, L19: delete 'continuously'

P2, L19: change to 'With automatic field instruments used widely in national weather operations,'

P2, L20-21: delete 'which are necessary in practical work'

P2, L24: 'tested this function using precipitation measurements from Norway and the US, and proposed…'

**2. Materials and Methods**

P3, L16: change to 'Chinese national precipitation gauges'

P3, L23: change 'eliminate' to 'mitigate'

P3, L29-31: change to 'The precipitation dataset was obtained from two sources: the recorded values from the TRwS$_{SA}$ and the manually observed values from the CSPG$_{DFIR}$.'

P4, L5: change 'we did not enable the heating options' to 'heating was not enabled'

P4, L10-13: change to 'Before calculating the TRwS$_{SA}$ precipitation amounts, the recorded values were subject to a manual quality control and filtering process. First, 30-min precipitation data were removed if the corresponding humidity was less than 50%. Second, 30-min precipitation data were removed if the corresponding sunshine duration was equal to 0.5 h.'

P4, L20: insert 'from the CSPG$_{DFIR}$' after 'reference precipitation'

P4, L22: delete 'to some extent'

P4, L27: change 'far away to each other' to 'far away from each other'

P4, L28: delete 'like' and pluralize 'aerodynamic effects'

P4, L32: change 'effectively prevent wind effect' to 'effectively mitigate wind effects'

P5, L4: change 'For $\Delta P_t$, it was assigned a value…' to '$\Delta P_t$ was assigned a value…'

P5, L5: change to 'TRwS$_{SA}$ measurements were considered to be subject to both specific errors and systematic errors.'

P5, L12: change 'are usually applied' to 'is usually applied'

P5, L13: change 'it seems difficult' to 'it is difficult'

P5, L14: change 'large amounts of data and various sites' to 'precipitation datasets from sites in different climate regimes'

P5, L22: change 'was aim' to 'aimed'

P5, L23: change 'means that there are no specific errors' to 'means that specific errors for the TRwS$_{SA}$ are not considered.'

P5, L24-25: change to '… Eq. (2); however, the quantification of specific errors for the TRwS$_{SA}$ is difficult. Therefore, Eqs. (1), (2) and (3) were applied to correct the TRwS$_{SA}$ measurements, ignoring specific errors.'

**3. Results and Discussion**

P6, L1: change 'Especially between' to 'Between'

P6, L7: delete 'obviously'

P6, L15-16: change to 'For snowfall events, wind effects will reduce the amount of snow reaching the TRwS$_{SA}$ gauge orifice.

P6, L16-17: change to 'Given that the orifice heating option was not enabled, snow accumulated on the gauge orifice may sublimate before melting and falling into the bucket.'

P7, L21-22: change 'It is obvious that these scatters disperse a lot on the plots' to 'Significant scatter is evident in both plots.'

P7, L31-32: change to 'The adjustment function in Eq. (3) is fit to the TRwS$_{SA}$ catch ratio as a function of both gauge-height wind speed and air temperature in Fig. 4.'

P8, L1-2: change to '…the corrected CRs is 0.01; this low value is not surprising, given the significant scatter observed in Fig. 4.'

P9, L14: consider changing to: 'As shown in Fig. 1, the shapes of the TRwS and CSPG are different: the TRwS is wider at the bottom, and more narrow at the top, with a 'shoulder' in between the two diameter portions; the CSPG is a cylinder of constant diameter.'

**Figures and tables**

P19, L4-5: change to '...and reference gauge, with 0.3 < CR < 1.7, were included.'

P21, L5-6: same as above

P21, L3: symbol for degrees Celsius looks strange

P25, L3: same as above

Table 2, 3: change 'Obse' to 'Obs'

P20 and 22, L4: change to… 'and in which 0.3 < CR < 1.7, were included.'

---

## Author Comment (AC3) · 11 Aug 2017

**Reply to the Editor**

Thanks very much for Dr. Michael Earle to review the revised manuscript; below we give the reply to the comments.

**Comments:**

**Title**

1. Propose changing title to 'Correcting precipitation measurements from MPS TRwS204 automatic weighing gauges in the Qilian Mountains, China'

**Reply:**

We agree with Dr. Michael Earle's proposal and we will make changes in the next revision.

**Abstract:**

1.The abstract should be brief, but still needs to introduce necessary background material. The gauges included in the study must be introduced (e.g. MPS TRwS204 is an automatic weighing gauge, CSPG is the manual Chinese Standard Precipitation Gauge) and the 'existing adjustment function' should be elaborated upon. Further, it is stated that 'deriving adjustment algorithms has become a top priority,' but there is no mention of what these algorithms are adjusting for (e.g. wind‐induced undercatch of precipitation). The abstract should be revised to include the above points.

**Reply:**

We agree with Dr. Michael Earle that the abstract needs to introduce necessary background material, and we will make modifications in combination with the proposed technical revisions.

2. When considering the results after adjustment, the following statements are made: 'It seems that the adjustment function is more appropriate to correct the snowfall measurements than rainfall and sleet measurements for this dataset.' This makes sense given the results presented, but only considers the average loss relative to the reference. What about the Root Mean Square Error? What about the Bias? It is stated that 'Overall, the results of the correction are not ideal,' but this statement is based only on the average loss, which could be impacted by a small number of events with larger losses relative to the reference. The assessment approach should be expanded, as will be discussed further in subsequent comments.

**Reply:**

We agree with Dr. Michael Earle that the statements were not very serious and there was no comprehensive consideration. We will expand the assessment approach in the next revision.

3. It is stated that 'so many factors seem to affect the differences between measurements,' but only two factors are noted (orifice area and wind profile). Other contributing factors should be described, or this sentence should be reformulated.

**Reply:**

We are going to change it to 'factors that orifice area, wind profile and random errors seem to affect the difference between measurements of or $CSPG_{SA}$ and $TRwS_{SA}$.'

4. The final sentence in the Abstract would be much stronger and more broadly applicable as 'These types of errors must be considered when correcting precipitation measurement errors for different gauge types and configurations.'

**Reply:**

We agree with Dr. Michael Earle, and we will change it.

**A) Introduction**

1. What is meant by "false" precipitation? (P2, L12)

**Reply:**

According to Bogdanova et al. (2002), "false" precipitation means that snow raised from the surface of the snow cover and caught by the gauge during blowing snow of blizzard.

2. It is stated that 'the transition from manual to automatic measurements was highly encouraged' by the SPICE IOC (P2, L25‐26) – can you please elaborate on this? If I recall correctly, SPICE was organized in response to the transition to automation, not to advance or recommend this transition.

**Reply:**

It could be that I misunderstood, and I will delete this sentence.

3. The Introduction transitions abruptly from a discussion of errors and adjustment functions to a discussion of manual vs. automated measurements (P2, L25‐32). This information is valuable, but seems out of place here. The biases in gauge measurements are assessed relative to reference measurements; historically (e.g. the first WMO Solid Precipitation Intercomparison), the reference measurements were manual measurements using the DFIR. I suggest that the authors revise the Introduction to first describe these biases (they are presently introduced without context), then describe known biases (e.g. for automated vs. manual measurements), and then get into the different errors/contributing factors and adjustment functions. In short, the Introduction should establish the context for interpreting the results that will be presented, and should flow logically from topic to topic.

**Reply:**

We agree with Dr. Michael Earle that the logic of this section is not very smooth, and we will consider the suggestion offered by Dr. Michael Earle and revised the introduction.

4. The transition from a discussion of biases between automatic and manual measurements to the statement 'Thus, intercomparisons at different sites around the world should be conducted to the test the performance of the automatic system and correct the precipitation measurements' is confusing. Why would testing at different sites around the world be helpful? It is difficult to follow the logic of this section, as currently presented.

**Reply:**

We agree with Dr. Michael Earle that it transit abruptly in this section, and we will revise this section. The sentence "Thus, intercomparisons at different sites around the world should be conducted to the test the performance of the automatic system and correct the precipitation measurements" will be deleted.

5. While it is true that the SPICE intercomparison sites could have their own measurement objectives, it is not necessary to state this here (P3, L1‐2), as the Qilian Mountains site was not a formal intercomparison site.

**Reply:**

We agree with Dr. Michael Earle, and this sentence will be deleted.

6. Which existing adjustment function? (P3, L7‑8)

**Reply:**

It refers to the adjustment function from Kochendorfer et al. (2017). We will revise this sentence.

**B) Materials and Methods**

1. Is the DFAR configuration (with TRwS204 gauge) at the Qilian Mountains site used in the analysis? If not – which I believe to be the case – the DFAR configuration does not need to be introduced and discussed on P3, L23‑27 (i.e. these two sentences can be deleted).

**Reply:**

The DFAR configuration (with TRwS204 gauge) at the Qilian station was not used in this study, and we agree with Dr. Michael Earle to delete these two sentences.

2. In Figure 1, it appears that the single‑Alter shield slats on the $TRwS_{SA}$ (Figure 1b) are installed differently than those on the $CSPG_{SA}$ (Figure 1c). Those on the $TRwS_{SA}$ are oriented with the flat side of the slat toward the gauge (correct), while those on the $CSPG_{SA}$ are oriented with the flat side of the slat away from the gauge (incorrect). The two shield configurations are therefore not identical. The location of the centre of mass and distribution of the slat surface area will be different in each case, impacting how the slats respond to a given wind speed. The shields being identical is an important assumption in the assessment, so this difference should be noted in the manuscript.

**Reply:**

First, we must admit that such mistake did occur during installation. As far as I know, there was no professional guidance on early installation so that we didn't notice this problem timely. The right single-Alter shield slats have been replaced in June 12, 2016. This difference will be noted in the manuscript.

[Figure]

3. How were the $TRwS_{SA}$ measurements adjusted to match the diameter of the CSPG? You indicate that the manufacturer changed a setting (P4, L10), but additional details would be helpful.

**Reply:**

The transfer function for TRwS to match the diameter of the CSPG is:

P2 = (P1/200cm$^2$)*314 cm$^2$,

where P2 refers to the converted precipitation of TRwS and P1 refers to the original precipitation of TRwS. This work was done by the instrument company.

4. In the manual quality control and filtering process, are you referring to the mean 30 min humidity? The details should be provided to guide those who may want to use a similar procedure. Also, it makes sense to remove precipitation during clear sky periods (sunshine duration = 0.5 h), but the threshold duration value is not very strict. For example, if the sunshine duration was 0.49 h, would the precipitation data be included in the analysis?

**Reply:**

We agree with Dr. Michael Earle and also feel sorry that we didn't make this section clear. We will change theses sentences to 'First, 30-min precipitation data were removed if the corresponding 30-min mean humidity was less than 50%. Second, 30-min precipitation data were removed if the corresponding 30-min sunshine duration was strictly equal to 0.5 h.'

5. It is stated that TRwS$_{SA}$ precipitation 'was compared with the reference precipitation to investigate the performance of the TRwS$_{SA}$ and correct its measurements.' This makes it sound like the comparison with the reference corrects the TRwS$_{SA}$ measurements, which is not the case. Perhaps it would be clearer to state simply that the TRwS$_{SA}$ measurements were assessed relative to reference precipitation measurements from the CSPG$_{DFIR}$? I don't know if it is necessary to mention the adjusted/corrected measurements and their assessment at this point.

**Reply:**

We agree with Dr. Michael Earle that this sentence is of some thoughtlessness, we will change it to 'Precipitation recorded by the TRwS$_{SA}$ was assessed relative to reference precipitation from the CSPG$_{DFIR}$.'

6. Again, no information is provided to indicate what is meant by "false" precipitation (P4, L30).

**Reply:**

We will change it to 'no "false" precipitation caused by blowing snow flux into the gauge (Bogdanova et al., 2002)'.

7. The reasons for not deriving transfer/adjustment functions from the experimental dataset and motivation for using the transfer function developed by Kochendorfer et al. are not clearly articulated. It is stated that 'it seems difficult to derive a valid and robust transfer function for TRwS$_{SA}$ using the dataset at this site during the experimental period.' Why is this difficult? What are the limitations of the dataset? Several important points were raised during the previous review stage and discussion, which should be reflected in the manuscript.

**Reply:**

We agree with Dr. Michael Earle that the reason for deriving adjustment functions from the experimental dataset and motivation for using the transfer function developed by Kochendorfer et al. are not clearly articulated. We will revise this section as 'However, because of rather small precipitation data set (especially for sleet and snow events) and limited meteorological conditions (lower mean wind speed) at this site during the experiment period, it is difficult to derive a valid and robust transfer function for TRwS$_{SA}$ using this data set.'

**C) Results and Discussion**

1. Numerous (20) 12‐h precipitation events were noted in which the $TRwS_{SA}$ did not report precipitation, but the $CSPG_{DFIR}$ reported precipitation. Four events were rain and sleet, during which the conditions were 'nothing special'; can you please reword and elaborate on this? 16 events were snowfall events, which were evidently characterized by lower temperatures. Do you have a theory to explain why this may have been the case? What were the characteristic wind speeds? For example, if the precipitation was light and the wind speeds were higher, it would not be surprising if the single Alter shielded gauge missed the event. Did the $CSPG_{SA}$ report precipitation during these events?

**Reply:**

This sentence "For rainfall and sleet events, …… were nothing special" should be revised or be deleted, because few rainfall and sleet events were missed by $TRwS_{SA}$, it is hard to find features of environmental conditions. The theory to explain why these 16 snowfall events were evidently characterized by lower temperatures was investigated in the next paragraph. 12-h mean wind speeds at gauge height varied from 0.4 to 1.7 m s$^{-1}$ for these 16 snowfall events, and the distribution was uniform. As presented in P5, L29 that "During the experiment, 304 precipitation events (12-h scale) were measured by $CSPG_{DFIR}$ (also by $CSPG_{SA}$)", the $CSPG_{SA}$ reported precipitation during these events.

Another important concern is whether the 12‐h conditions are representative of the conditions during which precipitation actually occurred during a given 12‐h period. There's not necessarily a better way that you could have addressed the conditions, but the representativeness of conditions is an important point that must be noted.

**Reply:**

In fact, in the section "Losses of $TRwS_{SA}$ Relative to Reference Precipitation", we included 304 precipitation events (12-h scale) which contain these 20 events missed by $TRwS_{SA}$. Because of no precipitation recorded by $TRwS_{SA}$ for these 20 events, it was hard to know specific precipitation time. Therefore, mean wind speed during precipitation for these 20 precipitation events can not be calculated. In this section, 12-h mean wind speeds were used for unity. In the sections "Adjustment for $TRwS_{SA}$ Measurements" and "Discussion on Specific Errors of $TRwS_{SA}$ Measurements", mean wind speeds at gauge height during precipitation at 12-h scale were used.

2. When discussing precipitation losses (e.g. P6, L9), it is important to specify what the losses are relative to (i.e. the reference configuration).

**Reply:**

We agree with Dr. Michael Earle, it may be changed to 'the losses of the recording electronic weight precipitation gauge relative to the standard non-recording Hellmann gauge.'

3. Snow accumulating on the orifice and sublimating is proposed as a loss mechanism for the $TRwS_{SA}$. Were any incidents observed during the experimental period in which snow accumulated on the orifice, or is this just a theory? Any accumulated snow could also prevent incident snowfall from entering the orifice and being measured – that is, capping of the gauge may occur – which would influence the assessment.

I would assume that the CSPG is not heated, either. Did you see any snow accumulation on the

CSPG? If so, is it included in the manual measurement, or is it removed prior to the measurement? Any differences in how precipitation accumulated on the gauge orifice is dealt with for the different gauge types could potentially impact the assessment.

Is there any other reason why accumulation/sublimation would be an issue for the $TRwS_{SA}$ and not the CSPG gauges? It is stated that accumulation/sublimation 'may explain most of the situations in which the $TRwS_{SA}$ recorded values of 0 for snowfall events,' but the reasons why this does not occur for the CSPG gauges are not discussed.

**Reply:**

Snow accumulating on the orifice and sublimating is just a theory. However, when I consulted the observer, he said that almost no accumulated snow on the orifice of TRwS and CSPG was observed in winter at this site. Obviously, I made a false theory. I will revise this part. Additionally, the manual CSPG is not heated.

4. Is data loss considered to be another precipitation loss mechanism for the TRwS gauge? If the gauge reports are based on the weight of accumulated precipitation, won't the reports following any periods of data loss include any precipitation accumulated in the bucket during the loss period?

**Reply:**

Yes, data loss was considered to be another precipitation loss mechanism for the TRwS gauge in this version of manuscript. The gauge reports were based on the weight of accumulated precipitation, however, the output of depth failed to work while the output of weight still worked for these 20 precipitation events. Because the output of weight worked normally, it can't be the problem of electricity shortage. We finally thought this may be the internal data processing problem.

5. The title of Table 1 indicates a comparison of events on a 12‑h scale; however, the notes indicate daily (24‑h) mean wind speeds and temperatures are presented. Is this correct? As noted above, the conditions during 12‑h periods are not necessarily representative of the conditions during which precipitation occurred. This issue will be more significant if conditions over 24‑h periods are used.

**Reply:**

I am sorry that I forgot to change "daily (24-h)" to "12-h", and the data in Table 1 are all at 12-h scale. Because 304 precipitation events (containing 20 events which missing precipitation data for $TRwS_{SA}$) were included in this section, we used 12-h mean meteorological variables for unity.

6. I would consider removing the statement 'It is obvious that the $CSPG_{SA}$ performed better than the $TRwS_{SA}$ during the experiment at this site' (P6, L32). This is a subjective statement. Alternatively, it could be changed to something like, 'It is evident that the $CSPG_{SA}$ collected more precipitation relative to the $TRwS_{SA}$ during the experiment at this site.'

**Reply:**

We agree with Dr. Michael Earle that this statement is subjective, we will change it.

At the low mean wind speeds experienced at the site over the experiment (Table 1), the differences between measurements from a single‑Alter shielded gauge and those from a

DFIR‑shielded gauge may not be significant. In this case, the $CSPG_{SA}$ and $CSPG_{DFIR}$ measurements would be expected to be very similar, and any differences relative to the $TRwS_{SA}$ could be attributed to systematic differences in the sampling, principle of operation, aerodynamic profile, etc. between the gauge types. These systematic differences are addressed in the $TRwS_{SA}/CSPG_{SA}$ comparison.

**Reply:**

We will add the analysis of the difference between $CSPG_{SA}$ and $TRwS_{SA}$ measurements. As Dr. Michael Earle stated, the aerodynamic differences between $CSPG_{SA}$ and $TRwS_{SA}$ would be expected to be small in such a small wind speed range. The difference between $CSPG_{SA}$ and $TRwS_{SA}$ measurements in this study may be attributed to difference in sampling, operating principle, etc. between the gauge types.

7. It is stated that 'the performance of the $CSPG_{SA}$ was more stable than that of the $TRwS_{SA}$' (P7, L13‑14), but this comparison does not acknowledge that the CSPG is a manual gauge (how can it be unstable?), or define what is meant by 'stable.' Is this statement made in reference to the lower standard deviation for the $CSPG_{SA}$? If so, when considering measurements over such long time scales (12‑h), does the standard deviation really say anything about noise or signal variability?

**Reply:**

We agree with Dr. Michael Earle that this statement "the performance of the $CSPG_{SA}$ was more stable than that of the $TRwS_{SA}$" is subjective. Here, I did make this statement in reference to the lower standard deviation for the $CSPG_{SA}$. As Dr. Michael Earle mentioned, measurements over such long tome scales (12-h) can not say anything about the noise or signal variability. I will delete this statement.

8. Aside from wind speed and temperature, which other 'meteorological variables have relationships with the catch ratio' (P7, L17)? Also, when noting that air temperature and wind speed are 'typically' applied in adjustment functions, it would be valuable to include supporting references.

**Reply:**

Aside from wind speed and temperature, humidity also has relationship with the catch ratio. I think it may be more better to change "Although most meteorological variables have…" to "Although several meteorological variables have…". We agree with Dr. Michael Earle that it would be valuable to include supporting references when noting that air temperature and wind speed are typically applied in adjustment function. It will be added in the next revised version.

9. The trends described for the catch ratio as a function of wind speed and temperature in Figure 3 (text on P7, L22‑24) are difficult to observe in the plots. Binning the results for each precipitation type (rain, sleet, snow) and plotting as box and whisker plots would provide a much clearer representation of the data trends, and are recommended to complement the scatter plots provided in Figure 3.

**Reply:**

We agree with Dr. Michael Earle's suggestion, and we will add this kind of pots in the next revised manuscript.

10. RMSE values are computed for observed catch ratios relative to adjusted values. It would be valuable to also include RMSE values for observed $TRwS_{SA}$ accumulation values relative to the adjusted values.

**Reply:**

We agree with Dr. Michael Earle's suggestion, and we will compute the RMSE values for observed $TRwS_{SA}$ accumulation values relative to the adjusted values in the next revised manuscript.

11. On page 8, the performance of the adjustment functions is assessed in terms of average precipitation losses in each precipitation type. This assessment should be complemented by RMSE and bias values for the observed $TRwS_{SA}$ accumulation values relative to the adjusted values. In the work of Kochendorfer et al., it was found that the adjustments improved the bias in value relative to the reference, and had less of an impact on the RMSE; it would be interesting to see if similar trends are observed using this experimental dataset.

**Reply:**

We agree with Dr. Michael Earle's suggestion, and we will add the content of analysis of RMSE and bias values for the observed $TRwS_{SA}$ accumulation values relative to the adjusted values.

12. Can you please elaborate on the 'other possible errors' that may contribute to differences between $CSPG_{DFIR}$ and $TRwS_{SA}$ measurements (P9, L4 – 5)?

**Reply:**

Errors caused by different aerodynamic profiles and orifice areas of TRwS and CSPG, and random errors.

13. Do you think that the similarity of the mean absolute differences between the $CSPG_{SA}$ and $TRwS_{SA}$ in all precip types (P9, L10) may indicate a systematic difference in measurements between these gauge configurations? That is, does the combined influence of errors specific to the $TRwS_{SA}$ result in a systematic offset in measurements relative to the CSPG?

**Reply:**

We agree with Dr. Michael Earle that the similarity of the absolute differences between the $CSPG_{SA}$ and $TRwS_{SA}$ in all precipitation types may indicate the difference in measurements between these gauge configurations. It may also relate to the most low mean wind speed during precipitation. Because the corrected $CSPG_{DFIR}$ measurements were regarded as reference precipitation, we prefer to believe that the combined influence of errors specific to $TRwS_{SA}$ mainly result in these differences.

14. In Figure 5, the cumulative sums of precipitation accumulations are plotted as a function of event number – this is effectively a time series of the total accumulated precipitation for each gauge at the end of each event. It would be far more instructive to plot the individual event accumulations (accumulation at end of 12 h period minus the accumulation at the start of that 12 h period) for the $TRwS_{SA}$ vs. those for the $CSPG_{SA}$ (or vice versa) as a scatter plot. That way, each event can be compared independently, irrespective of the precipitation accumulated in previous events. A 1:1 line (indicating perfect agreement between the gauges) can be added to illustrate the accumulation trends.

**Reply:**

We agree with Dr. Michael Earle, and I had some misconceptions here before. I will change this plot to scatter plots of TRwS$_{SA}$ vs. CSPG$_{SA}$ event measurements.

15. As presented, the results in Figure 6 are difficult to interpret. I recommend generating histograms of the differences between the CSPG$_{SA}$ and TRwS$_{SA}$ measurements for each precipitation type to more clearly illustrate the magnitude of differences between the gauges, and how those differences are distributed. Alternatively, the TRwS$_{SA}$ vs. CSPG$_{SA}$ scatter plots (see comment above) could be modified to include only points events with mean wind speeds < 1 m/s.

**Reply:**

We agree with Dr. Michael Earle that it is a little hard to see the mean absolute difference for each precipitation type since I combined all the precipitation events for gauge-height wind speed below 1 m s$^{-1}$. We will plot them separately in the next version.

16. As noted above, the TRwS$_{SA}$ and CSPG$_{SA}$ technically have the same shields, but the orientation of the slats is different (Fig. 1). This potentially complicating factor should be noted in the relevant discussion in Section 3.3.

**Reply:**

We agree with Dr. Michael Earle, and this issue will be elaborated in the relevant discussion in Section 3.3.

17. It is interesting that limiting the CSPG$_{SA}$/TRwS$_{SA}$ comparison to wind speeds below 1 m/s results in almost the same mean absolute differences between the measured values as observed for the full wind speed range. I wonder if this is a reflection of the full wind speed range being low (at least in terms of the mean wind speeds in Table 1), a reflection of the mean wind speeds for 12‑h periods not being representative, or an indication of differences/errors not related to the aerodynamic profile, as proposed.

**Reply:**

Here, we need to restate that the gauge-height mean wind speeds during precipitation at 12-h scale were used in section 3.2 and 3.3. As shown in Fig.3, the full mean wind speed during precipitation is about in the range of 0 to 3 m s$^{-1}$. However, the most wind speeds were low. In my opinion, such phenomena may be a reflection of the full wind speed range being low as Dr. Michael Earle stated. In this way, the difference/errors caused by aerodynamic effect can be small from the results for these two wind speed ranges. This should be stated in the manuscript, and we will elaborate on it.

18. The trends indicated in Figure 7 do not appear to be conclusive for sleet and snow, given the significant scatter and small numbers of events relative to rain (I expect the R$^2$ correlation values would be low). This caveat should be noted when making the statement, 'it may be inferred that the amount of precipitation mainly affects the specific errors of TRwS$_{SA}$ during the experiment at this site.'

**Reply:**

We will pay attention to this problem considering small numbers of sleet and snow events, and this kind of statement will be revised.

D) Conclusions

1. As noted above, I don't think that the standard deviation of losses is a representation of measurement stability; the losses are larger for the $TRwS_{SA}$, so the standard deviation of loss values will also be larger.

**Reply:**

We agree with Dr. Michael Earle that it may be not appropriate to compare the stability of TRwS and CSPG because of different operational methods. Additionally, as Dr. Michael Earle pointed, measurements over such long tome scales (12-h) can not say anything about the noise or signal variability. I have different opinions about the statement that 'the losses are larger for the $TRwS_{SA}$, so the standard deviation of loss values will also be larger'. When the differences were small between the losses for each event, the standard deviation of loss values can be small even if the losses are larger.

2. How is it 'clear' that the 'worse correction for $TRwS_{SA}$ measurements should not be attributed to the limited meteorological conditions and precipitation data during the experiment'? I don't believe that these points were addressed in the manuscript; an earlier comment requested more details in this regard.

**Reply:**

This has been addressed in P8, L20. In the third paragraph in P8, to verify whether the limited meteorological conditions would contribute to the poor correction for $TRwS_{SA}$ measurements, the same adjustment function was used to $CSPG_{SA}$ measurements. From the result, different correction effects occurred. Because the $TRwS_{SA}$ and $CSPG_{SA}$ experienced the same meteorological conditions, the presented correction effect should not be attributed to the limited meteorological conditions. We will revise this part and add more details.

3. I don't know if you are in a position to say that the adjustments are 'better' or 'worse' when the changes in loss values after adjustment for both gauges are within 0.2 mm for all precip types. You indicate that the measurement uncertainty is ignored (P10, L27), but what is the estimated uncertainty for each gauge type? Perhaps more important, you are using wind speeds over 12‑h periods for the adjustment that do not necessarily reflect the conditions during which precipitation actually occurred, which will impart significant uncertainty on the adjusted values. I think it is OK to state the results obtained, but any broader application of these results should be treated with extreme caution.

**Reply:**

Maybe the content about correction for $TRwS_{SA}$ and $CSPG_{SA}$ which put in the conclusion is not very appropriate. As presented in P8, the change of measurements of $TRwS_{SA}$ for all precipitation types after correction is -33.3 mm (in L10), while the change of measurements of $CSPG_{SA}$ for all precipitation types after correction is 20.2 mm (in L19). The correction for $CSPG_{SA}$ measurements seems to be better than correction for $TRwS_{SA}$ measurements. We will also combine the RMSE and bias values to discuss it. We agree with Dr. Michael Earle that any broader application of these results should be treated with extreme caution. We will check our wording again. Additionally, we agree with Dr. Michael Earle that the estimated uncertainty may be from the unrepresentative mean wind speed during precipitation at 12-h scale. I think that the sample size will also cause the estimated uncertainty.

4. The last two sentences on page 11 are very difficult to follow. Can you please rephrase these, including relevant results from the study, if possible, to demonstrate your points?

**Reply:**

In the work of Kochendorfer et al. (2017), the adjustment function Eq. (3) was derived for correcting wind-induced loss for single Alter shielded Geonor gauge, and its reference precipitation was the measurements of DFAR shielded Geonor gauge. In this study, the same adjustment function was used for $TRwS_{SA}$ measurements, but the reference precipitation was the corrected measurements from DFIR shielded manual gauge (CSPG). Because in addition to the wind-induced loss needing correction, the specific errors for $TRwS_{SA}$ were thought may also require to be corrected in this study. After applying the adjustment Eq. (3), we can find that it was no completely calibrated for $TRwS_{SA}$ measurements. This result seems to be consistent with the assumption we made before, so it can be reasonable.

**References:**

Bogdanova, E. G., Ilyin, B. M., and Dragomilova, I. V.: Application of a comprehensive bias-correction model to precipitation measured at Russian North Pole drifting stations, J. Hydrometeorol., 3, 700–713, doi: 10.1175/1525-7541(2002)003<0700: AOACBC>2.0.CO; 2, 2002.

Kochendorfer, J., Rasmussen, R., Wolff, M., Baker, B., Hall, M. E., Meyers, T., Landolt, S., Jachcik, A., Isaksen, K., Brækkan, R., Leeper, R.: The quantification and correction of wind-induced precipitation measurement errors, Hydrol. Earth Syst. Sci., 21, 1973–1989, doi: 10.5194/hess-21-1973-2017, 2017.

**Finally, thanks again for Dr. Michael Earle spending his valuable time pointing out the problems and offering suggestions for this manuscript.**